# StyleDistillation: A New Insight of Image Style Enables Personalized Aesthetic Manipulation

Yuxin Wang [1 2]  Xiaoyu Geng [1 2]  Yuke Li [1 2 †]  Zheng Wang [1 2 †]

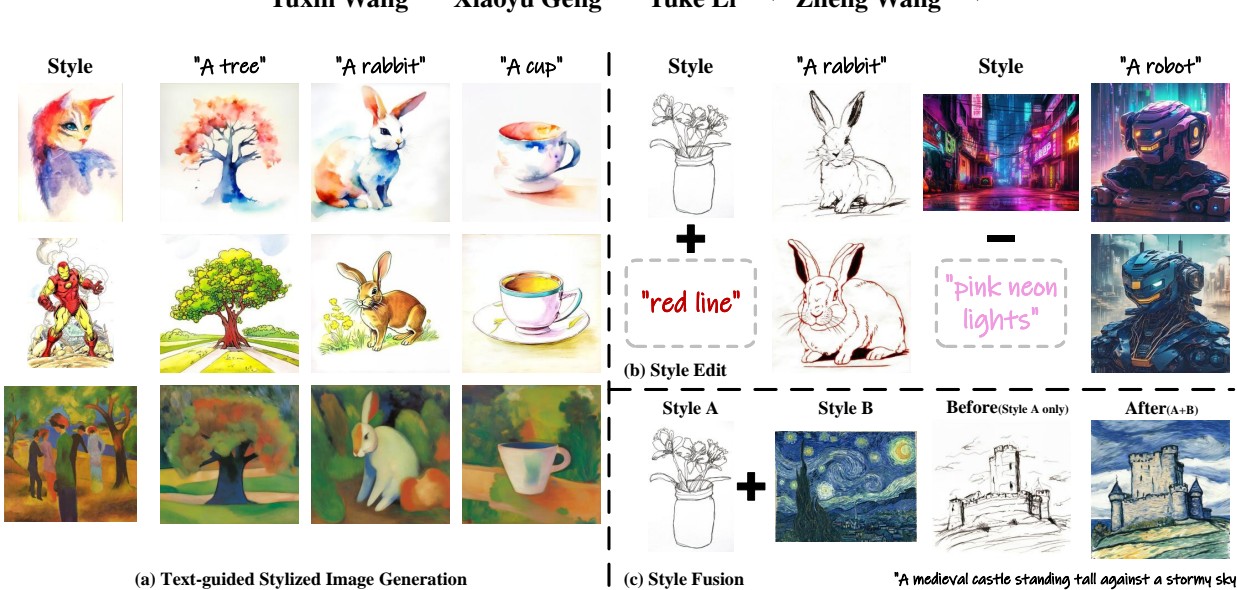

*Figure 1.* StyleDistillation enables (a) **text-guided stylized image generation** with high style fidelity and prompt alignment, (b) **style editing** via text-driven element modification, and (c) **multi-style fusion** for novel visual effects.

## Abstract

Text-guided stylized image generation has yielded promising advances by leveraging the powerful capabilities of text-to-image diffusion models. However, the inherent coupling of style and content information within the reference image presents a significant challenge. To address this, we propose StyleDistillation, a novel approach grounded in two key observations about the CLIP embedding space from a style perspective. By leveraging a lightweight StyleDistiller module, combined with carefully designed optimization objectives based on geometric and semantic priors, we can extract fine-grained style representation from the reference image. Additionally,

we introduce a Prompt Alignment Enhancement mechanism during inference, which significantly improves the control that text prompts exert over the generated images. Extensive experiments demonstrate that our method achieves outstanding performance in both style reproduction and prompt alignment. Furthermore, StyleDistillation supports various personalized operations, including style editing and style fusion, highlighting its substantial potential for diverse applications.

## 1. Introduction

Text-guided stylized image generation, or text-driven style transfer, leverages the power of text-to-image (T2I) models (*e.g.*, Stable Diffusion) (Chang et al., 2023; Rombach et al., 2022; Podell et al., 2024) to blend the style of a reference image with the content specified by a text prompt. In this work, we focus on the practical yet challenging *single-reference* scenario, where a user typically provides only one image to define a personalized style, and aim to synthesize images that faithfully reproduce its fine-grained style while maintaining strong content alignment with the text prompt.

However, the inherent coupling of style and content infor-

†Corresponding author. [1]National Engineering Research Center for Multimedia Software, Institute of Artificial Intelligence, School of Computer Science, Wuhan University [2]Hubei Key Laboratory of Multimedia and Network Communication Engineering. Correspondence to: Yuke Li <sunfreshing@whu.edu.cn>, Zheng Wang <wangzwhu@whu.edu.cn>.

*Proceedings of the 43rd International Conference on Machine Learning*, Seoul, South Korea. PMLR 306, 2026. Copyright 2026 by the author(s).

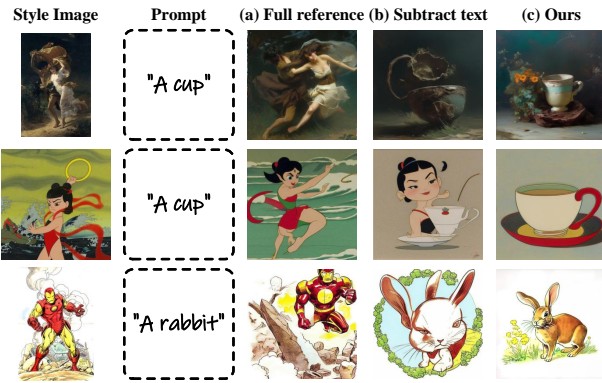

*Figure 2.* **Different forms of style conditioning during image generation.** More results are presented in the Appendix C.1.

mation within the reference image presents a significant challenge. In our setting, all personalized style information comes from the user-provided reference image, so a natural idea is to directly treat it as a style constraint during generation. Using the pre-trained image encoder of CLIP (Radford et al., 2021; Ye et al., 2023), whose text encoder is already used in diffusion models (Rombach et al., 2022; Podell et al., 2024) to encode text prompts, we can encode the reference image into a feature embedding and inject it into the diffusion model together with the text prompt. In practice, however, style-irrelevant content information in the reference image severely interferes with the generation results. As illustrated in column (a) of Figure 2, the character from the left reference image inappropriately appears in nearly every case. Such interference distorts the structure of the generated images, making it difficult to accurately reproduce the reference style (*e.g.*, in the "rabbit" case), and at the same time makes it nearly impossible for the images to align with the prompt text in terms of content.

Therefore, the key objective becomes to extract fine-grained style representation from the reference image while effectively filtering out content interference. Leveraging the capability of CLIP to encode both text and images into a shared embedding space, a straightforward approach (Wang et al., 2024a) is to generate a textual description of the reference image's content and subtract it from the image embedding to isolate the style. However, as shown in column (b) of Figure 2, such naive subtraction not only fails to sufficiently filter out content interference (*e.g.*, in the "cup" case of the second row), but more critically, it also degrades the reference style details (*e.g.*, losing lighting nuances in the first row and color fidelity in the second and third rows). We argue that these limitations stem from the fact that simple subtraction relies on the fragile assumption of perfect text-visual alignment, while simultaneously overlooking the intrinsic geometric structure of style within the embedding space. This prompted us to re-examine the CLIP space from a style perspective, leading to two key observations:

1. In the CLIP embedding space, the effect of a given style on image embeddings is highly consistent across diverse contents, forming well-structured clusters, which provides a robust geometric prior for extracting fine-grained style representation.

2. Textual descriptions are far from perfectly aligned with the visual elements of a reference image, so treating the text description as an exact "content component" for subtraction is fragile in principle. Instead, it is more reliable to use the description as semantic guidance for content separation.

Motivated by these insights, we introduce StyleDistillation, a novel framework for distilling a pure style representation from the reference image. Specifically, its core StyleDistiller module learns a style representation $y_s$ together with an auxiliary content-related representation $y_c$ from the reference image embedding $f_I^s$. The module first utilizes textual descriptions that separately detail the content and style of the reference image to obtain an initial pair of representations from $f_I^s$, and further refines them through a set of carefully designed optimization objectives to produce the final style representation. Based on Observation 2, we employ the content description as semantic guidance to neutralize content interference by constraining the semantic direction of $y_c$. With the content effectively controlled, we then leverage Observation 1 to use the spatial position of $f_I^s$ as a geometric prior to anchor $y_s$, enabling it to capture fine-grained stylistic characteristics. We further introduce orthogonality and reconstruction constraints to govern the relationship between $y_s$ and $y_c$, ensuring that these two complementary priors work in synergy for a high-quality style representation. This insight-driven separation yields images that not only exhibit high fidelity to the reference style but also maintain strong alignment with the text prompt. To further enhance this alignment, we introduce a lightweight Prompt Alignment Enhancement mechanism during inference. The overall effectiveness of our method unlocks new possibilities for fine-grained, personalized aesthetic manipulation, including fusing multiple reference styles and directly editing a style's attributes with textual instructions. Our main contributions are as follows:

- We propose StyleDistillation, a novel framework for single-reference, text-guided stylized image generation. Its core StyleDistiller module, together with a lightweight Prompt Alignment Enhancement mechanism, enables images with both high style fidelity and strong prompt alignment.

- To obtain a high-quality style representation, we first make two key observations about the CLIP embedding space and abstract them into geometric and semantic priors. Building on these priors, we design a set of cooperative optimization objectives that effectively

extract style while suppressing content interference within the StyleDistiller module.

- Our framework unlocks novel capabilities for personalized aesthetic manipulation, enabling versatile applications like style fusion and style editing, the latter of which, to the best of our knowledge, remains underexplored in existing stylization methods.

## 2. Related Work

**Stylized Image Generation with T2I Models.** With advancements in T2I models (Rombach et al., 2022; Podell et al., 2024; Chang et al., 2023), stylized image generation has gained attention. Common approaches involve fine-tuning diffusion models. Textual inversion (Gal et al., 2022; Ahn et al., 2024) use learnable embeddings to capture style but struggle with fine details. Extending these embeddings to neural networks (Zhang et al., 2023; 2024d;c) improves expressiveness but requires style-specific training on large datasets. Methods like DreamBooth (Ruiz et al., 2023; Park et al., 2024) and LORA (Hu et al., 2022; Wang et al., 2025; Sohn et al., 2023; Zhang et al., 2024a; Liu et al., 2024; Choi et al., 2024) optimize U-Net parameters for better style capture but suffer from overfitting and content leakage. StyleAligned (Hertz et al., 2024) and (Zhou et al., 2025) extract style via self-attention, but increase inference costs and may reduce the diversity of generated images. Adapter-based methods like IP-Adapter (Ye et al., 2023) use pre-trained CLIP models to encode image features and inject them via a lightweight adapter, ensuring visual consistency. However, entanglement of style and content in reference features leads to content leakage and poor prompt alignment. Recent works (Chen et al., 2024; Huang et al., 2025; Borse et al., 2025; Qi et al., 2024; Li et al., 2024) including StyleShot (Gao et al., 2024) and CSGO (Xing et al., 2024) train style extractors on large datasets but are costly and show limited generalization to unseen styles. InstantStyle (Wang et al., 2024a;b) attempts to separate style by subtracting content descriptions, but this fails to effectively remove content and meanwhile corrupts the style information. In contrast, our approach distills high-quality style representation from a single reference image without style-specific datasets or retraining the diffusion backbone. This representation plugs into existing pipelines, providing a better balance between style fidelity and prompt alignment.

**CLIP Embeddings in Image Generation.** Diffusion models (Rombach et al., 2022; Podell et al., 2024) commonly utilize CLIP (Radford et al., 2021) encoders to map text and image concepts into a shared embedding space for conditional generation. This has prompted extensive discussions on the role of CLIP embeddings in image generation. Studies like CONCEPTOR (Chefer et al., 2024) and SpLiCE (Bhalla et al., 2024) decompose dense CLIP representations

into more interpretable concept components, highlight the composability of CLIP embeddings. Leveraging this, ToMe (Hu et al., 2024) constructs composite embeddings by combining CLIP text embeddings of different concepts, thereby addressing the semantic binding problem during image generation. SADis (Qin et al., 2025) attempts to independently control the color and texture of the generated image. IP-Composer (Dorfman et al., 2025) focuses on integrating multiple personalized concepts. In contrast, we analyze the CLIP embedding space from a style perspective and observe that style directions are largely consistent across content, while text is imperfectly aligned with visual components, jointly motivating our StyleDistillation framework.

## 3. Method

**Preliminary: Latent Diffusion Models.** Latent Diffusion Models, represented by Stable Diffusion (Rombach et al., 2022; Podell et al., 2024), are widely used for text-to-image generation. In Stable Diffusion, an image is encoded into a latent $z$, and Gaussian noise $\epsilon$ is added at time step $t$ to obtain the noisy latent representation $z_t$. The U-Net $\epsilon_\theta$ is trained to predict the noise by minimizing the loss:

$$\mathcal{L}(\theta) = \mathbb{E}_{z,c,t,\epsilon\sim\mathcal{N}(0,1)}\left[\|\epsilon - \epsilon_\theta(z_t, t, c)\|_2^2\right], \quad (1)$$

where $c$ represents the conditioning inputs, typically a text description of the image. Adapter-based methods like IP-Adapter (Ye et al., 2023) further allow an image embedding $f_I$ to be used as an additional condition, so that the predicted noise of U-Net can be expressed as $\epsilon_\theta(z_t, t, c, f_I)$.

### 3.1. Task Setup

We study text-guided stylized image generation in the single-reference setting. Given a reference image $I_s$, we obtain its feature embedding $f_I^s = CLIP_i(I_s)$ using a pre-trained CLIP image encoder. Our goal is to learn a style representation $y_s$ from $f_I^s$ and use it, together with a text prompt $p$, to guide image generation in a text-to-image diffusion model at inference time. The generated images are expected to faithfully reproduce the style details defined by the reference image $I_s$ while remaining semantically aligned with the content described by $p$.

### 3.2. Motivation

To effectively extract style from image embeddings while filtering content interference, a natural question arises: how does style information manifest within the CLIP space?

We investigate this through an empirical analysis on the UnlearnCanvas (Zhang et al., 2024b) dataset, mapping 400 images with different content to 60 unique styles. We first examine whether the effect of style can be directly observed

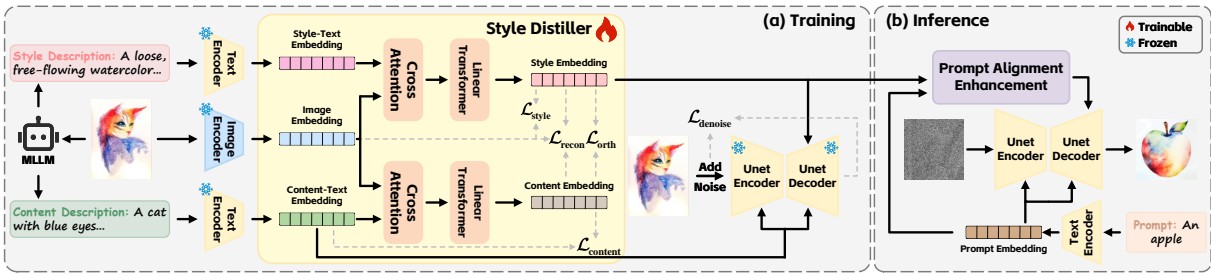

*Figure 3.* **An Overview of Our Method.** (a) Given a reference style image, we use the StyleDistiller module to extract style embedding with a carefully designed optimization objective. (b) During inference, we employ Prompt Alignment Enhancement to improve content alignment between generated images and text prompts.

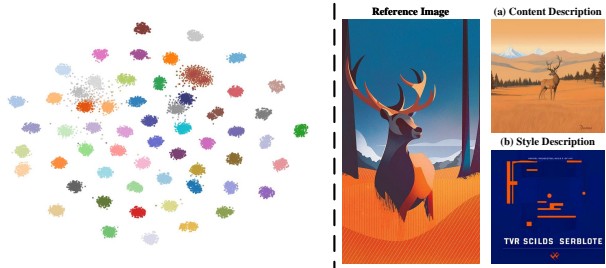

*Figure 4.* **Left:** T-SNE analysis for UnlearnCanvas (Zhang et al., 2024b) (colored by style). **Right:** Generation results based on content and style descriptions of the same reference image.

in the raw CLIP image embeddings. However, embeddings corresponding to the same style do not form well-separated groups and instead heavily overlap with those of other styles. By contrast, grouping by content yields much clearer separation: the average silhouette score (Rousseeuw, 1987), a measure of cluster separability, is about four times higher for content than for style (0.137 vs. 0.035). This suggests that content information plays the primary role in shaping the embedding, while the influence of style appears mainly as a weaker, content-coupled offset. A question then follows: does a style affect image embeddings in a content-dependent way or in a consistent manner across contents? We obtain style offsets by removing the projection of each embedding onto its content direction, where the latter is given by the mean embedding of images with the same content. As the t-SNE (Van der Maaten & Hinton, 2008) analysis in Figure 4 shows, these style offsets exhibit strong cross-content consistency: vectors corresponding to the same style are highly aligned in direction and form tight clusters. This yields Observation 1: in CLIP space, a style's influence on image embeddings manifests as a stylistic offset that is largely consistent across different contents. This finding inspires our core strategy: if content can be effectively controlled during optimization, this consistent geometric property can be leveraged to build a prior for style extraction.

However, controlling content is challenging in single-reference settings with only one image per style. We thus explore whether text can provide additional supervision. We use a multimodal large language model (MLLM) (Hurst et al., 2024) to generate separate content and style descriptions for a reference image and used them to guide image generation, as illustrated in Figure 4. This experiment reveals two key properties of textual descriptions. First, despite the style description being more detailed than the content one (on average about 3.5 times longer), the content description is far more effective at restoring the corresponding information of the reference image, while the style description largely fails. This indicates that text is more effective at capturing content semantics than stylistic details. Second, we observe that even the content-based reconstruction cannot perfectly match the visual details of the original image (e.g., the deer's position and expression, the distant mountains, and the trees), exposing an inherent misalignment between text and visuals. This directionally correct but detail-flawed characteristic forms our Observation 2: textual descriptions are unsuitable as an exact content component in CLIP space, but are well suited to serve as soft semantic guidance for constraining content in our framework. More experimental details are provided in the Appendix C.2.

## 3.3. StyleDistiller

To obtain the desired style representation $y_s$ from the reference image embedding $f_I^s$, we propose the StyleDistiller module. It takes $f_I^s$ as input, together with a pair of textual descriptions of the same image: a content description $d_c$ and a style description $d_s$, typically generated by an MLLM. Inside StyleDistiller, we design two parallel branches. While the main branch focuses on producing the style representation $y_s$, we introduce an auxiliary content branch that outputs $y_c$. This auxiliary representation is only used during training to make the control of content information more explicit and to help learn a cleaner style representation. Within these two branches, we leverage the semantic difference between $d_s$ and $d_c$ to process $f_I^s$ and obtain $y_s$ and $y_c$:

$$
\begin{aligned}
y_s &= W_s\big(\text{Attn}_{\text{sty}}(f_t^s, f_I^s)\big), \\
y_c &= W_c\big(\text{Attn}_{\text{con}}(f_t^c, f_I^s)\big),
\end{aligned}
\tag{2}
$$

where $f_t^s = CLIP_t(d_s)$ and $f_t^c = CLIP_t(d_c)$ are the CLIP text embeddings of $d_s$ and $d_c$, and $W_s$ and $W_c$ are linear mappings. $\text{Attn}(\cdot, \cdot)$ denotes cross-attention between text and image, computed as $\text{Attn}(f_t, f_I) = \text{softmax}\left(\frac{Q_t K_I^\top}{\sqrt{d_k}}\right) V_I$, where text embedding $f_t$ is used as the query and the image embedding $f_I$ as the key and value.

The above design provides an initial mechanism for extracting style information from the reference image, while the final quality of the style representation $y_s$ is largely determined by how StyleDistiller is trained. As a key component of our method, we therefore introduce a set of jointly designed training objectives, derived from the two observations in Section 3.2. These losses jointly optimize the style representation $y_s$ and the auxiliary content representation $y_c$, and regulate their interaction, ultimately yielding a high-quality style representation. Specifically, we define:

**Anchor Loss.** Our anchor loss consists of two complementary components, each responding to one of our key observations and providing directional guidance for the optimization of $y_s$ and $y_c$. Following Observation 2, textual descriptions cannot perfectly align with all visual elements of the reference image, we use the content description as semantic guidance to construct a content anchor for $y_c$. In CLIP space, the cosine similarity between embeddings is used to measure the similarity of their encoded information. Therefore, by constraining the cosine similarity between $y_c$ and $f_t^c$, we encourage $y_c$ to concentrate on content-related information. For the style representation $y_s$, the unreliability of style descriptions observed in Observation 2 makes a text-based anchor undesirable. At the same time, Observation 1 highlights that a stable style direction exists within $f_I^s$, but it is heavily obscured by content information. This leads to the following strategy: once the content-related information in $f_I^s$ is properly controlled during training, minimizing the Euclidean distance between $y_s$ and $f_I^s$ becomes an effective guide, compelling $y_s$ to align with the masked style direction. This control over content is jointly enforced by the orthogonality and reconstruction constraints, which we detail next. Accordingly, we define the anchor loss as:

$$\begin{aligned} \mathcal{L}_{\text{content}} &= 1 - \mathbb{E}\left[\text{ClipSim}(f_t^c, y_c)\right], \\ \mathcal{L}_{\text{style}} &= \mathbb{E}\left[\|y_s - f_I^s\|_2^2\right], \end{aligned} \quad (3)$$

where we define $\text{ClipSim}(\mathbf{a}, \mathbf{b}) = \max\left(0, \frac{\mathbf{a}^\top \mathbf{b}}{\|\mathbf{a}\| \cdot \|\mathbf{b}\|}\right)$, and the total anchor loss is $\mathcal{L}_{\text{anchor}} = \alpha \cdot \mathcal{L}_{\text{content}} + \beta \cdot \mathcal{L}_{\text{style}}$, with $\alpha$, $\beta$ denoting content and style anchor strengths.

**Orthogonality Loss.** While the anchor loss provides directional guidance for optimizing $y_s$ and $y_c$, it alone is insufficient to suppress content interference in the style representation $y_s$. To further encourage $y_c$ to focus on content and guide $y_s$ toward style information, we introduce an or-

thogonality constraint between the two representations. Intuitively, with $y_c$ anchored to content semantics, enforcing orthogonality naturally encourages $y_s$ to capture complementary style-related information. Concretely, minimizing this loss discourages high cosine similarity between $y_s$ and $y_c$ and pushes them toward orthogonality in CLIP space. The loss function is defined as: $\mathcal{L}_{\text{orth}} = \mathbb{E}\left[\text{ClipSim}(y_s, y_c)\right]$.

**Reconstruction Loss.** Beyond anchor and orthogonality constraints, we further introduce a reconstruction loss to encourage $y_s$ to fully capture the style information within $f_I^s$. Since there is no explicit ground-truth style representation to directly supervise $y_s$, we instead impose a global consistency constraint by encouraging a fused representation of $y_s$ and $y_c$ to remain close to the original image embedding $f_I^s$. To keep this constraint simple and stable, we construct the fused representation as a weighted combination of the L2-normalized versions of $y_s$ and $y_c$, denoted by $\tilde{y}_s$ and $\tilde{y}_c$: $\tilde{y}_{\text{fuse}} = \tilde{y}_s + \gamma \cdot \tilde{y}_c$, where $\gamma$ is a scalar coefficient controlling the contribution of $y_c$. The purpose of this loss is not perfect reconstruction, but to use this global constraint to drive $y_s$ to capture style information that is not already represented by $y_c$. To prevent $y_s$ from degenerating and losing stylistic details, we explicitly enforce $\gamma < 1$, thereby limiting the influence of $y_c$ and discouraging the fused representation from over-relying on $y_c$ during optimization. Section 4.3 details this design. Our reconstruction loss is defined as:

$$\mathcal{L}_{\text{recon}} = 1 - \mathbb{E}\left[\text{ClipSim}(\tilde{y}_{\text{fuse}}, f_I^s)\right], \quad (4)$$

**Denoise Loss.** The aforementioned optimization objectives in the CLIP embedding space impose fine-grained constraints on the style representation $y_s$. Additionally, we jointly train with a standard diffusion denoising loss, following Equation 1, to make $y_s$ better adapted to the underlying generation task. In our setting, we inject $y_s$ into the decoder side of the U-Net using IP-Adapter. Previous work (Wang et al., 2025) has shown that this part of the model is strongly correlates with the style of the generated images. We simultaneously inject $d_c$ into the model as a textual condition to encourage an effective division of labor during the denoising process, thereby enabling a better utilization of $y_s$ for rendering the visual style of the image. The predicted noise of U-Net can be expressed as $\epsilon_\theta(z_t, t, CLIP_t(d_c), y_s)$.

Finally, our optimization objective can be expressed as:

$$\mathcal{L}_{\text{Distillation}} = \mathcal{L}_{\text{anchor}} + \mathcal{L}_{\text{orth}} + \mathcal{L}_{\text{recon}} + \mathcal{L}_{\text{denoise}}. \quad (5)$$

Together, these objectives act in concert to distill from the reference image a high-quality style representation $y_s$ that effectively guides the diffusion-based generation.

### 3.4. Prompt Alignment Enhancement

After obtaining high-quality style representations $y_s$ via the StyleDistiller module, we leverage them during inference to guide the generation of images consistent with the reference style. However, when both $y_s$ and a text prompt are simultaneously injected as conditioning inputs, the model often tends to prioritize style, as $y_s$ typically contains richer visual information than textual prompts. This imbalance may lead to weakened content alignment between the generated images and their corresponding prompts. To address this challenge, we introduce a lightweight Prompt Alignment Enhancement (PAE) mechanism. Given a text prompt $p$, we use the CLIP text encoder to encode it into the CLIP embedding space, denoted as $f_t^p = CLIP_t(p)$, combine it with $y_s$ to obtain style-prompt embedding $y_s^p$, and inject it into the diffusion model, represented as:

$$y_s^p = y_s + \lambda \cdot f_t^p, \qquad (6)$$

where $\lambda$ controls the balance between style fidelity and prompt alignment. Furthermore, the ability of our method to extract purified style representations $y_s$, combined with insights from the PAE mechanism, motivates us to explore more advanced aesthetic manipulations. In particular, our framework enables flexible style fusion to synthesize entirely new artistic effects, and supports fine-grained, personalized style editing guided by textual instructions. These capabilities are demonstrated in Section 4.4. The overall algorithm of our method is given in Appendix B.

## 4. Experiments

### 4.1. Experimental Setups

**Evaluation.** To ensure a fair evaluation, we randomly selected 40 reference style images from Styleshot (Gao et al., 2024) and use GPT-4o to generate 30 text prompts of varying complexity to describe a diverse range of content. Each prompt produces 5 images, yielding 6,000 results per method. Following prior work, we separately evaluate the style reproduction capability and prompt alignment effectiveness of the generated images. The CSD score (Somepalli et al., 2024) is used to measure the style similarity between the generated results and reference images, while the CLIP-Text score (Radford et al., 2021) assesses the content alignment with the prompts and generated images.

**Composite Quality Score.** Since CSD and CLIP-Text focus on individual aspects, optimizations for one often impact the performance of the other, as shown in Figure 8. To explicitly reflect this trade-off, we propose the Composite Quality (CQ) score, which jointly considers style reproduction and prompt alignment in a single metric. By aggregating the two metrics, CQ facilitates direct comparison across methods

and provides a convenient reference for analyzing overall performance. Given the CSD and CLIP-Text scores of each method, the CQ score of the $i$-th method is defined as $CQ_i = \sqrt{\left(\frac{x_{sty}^i}{\sigma_{sty}}\right)^2 + \left(\frac{x_{text}^i}{\sigma_{text}}\right)^2}$, where $x_{sty}^i$ and $x_{text}^i$ represent the CSD and CLIP-Text score, respectively, and $\sigma$ denote the standard deviation of the respective metrics over all compared methods. Intuitively, for each method, CQ measures the Euclidean distance from the origin in a two-dimensional space whose axes are style and prompt performance, after normalizing each axis to account for the different scales of the two metrics. More details about the CQ score are provided in the Appendix D.

**Baselines and Implementation.** We select baselines based on task consistency, aiming to compare representative methods applicable to the same setting of single-reference, text-guided stylized image generation. To cover diverse technical paradigms in prior work, we include several state-of-the-art baselines, including InstantStyle (Wang et al., 2024a), CSGO (Xing et al., 2024), IP-Adapter (Ye et al., 2023), StyleShot (Gao et al., 2024), DEADiff (Qi et al., 2024), DreamStyler (Ahn et al., 2024), and StyleAlign (Hertz et al., 2024), using their official implementations. In addition, our method is built upon SDXL (Podell et al., 2024), leveraging the pre-trained IP-Adapter projectors (Ye et al., 2023) to inject style representations into the diffusion model. We use ChatGPT-4o to generate content and style descriptions for the reference images and train StyleDistiller with a learning rate of 0.001 for 300 steps, taking 150 seconds on a single NVIDIA RTX-4090 GPU. We set hyperparameters to $\gamma = 0.25$, $\alpha = 0.5$, $\beta = 0.1$. During inference, we use 50 timesteps with $\lambda = 0.35$.

### 4.2. Main Results

**Qualitative Comparisons.** Figure 5 presents a qualitative comparison with state-of-the-art methods. Since the content interference in the reference images was not filtered out, IP-Adapter (Ye et al., 2023) struggled to align with the prompt text effectively. InstantStyle's (Wang et al., 2024a) simple subtraction leads to a loss of style information, as seen in the "wolf" case, where the tone of the reference style is diminished. CSGO (Xing et al., 2024), Styleshot (Gao et al., 2024), and DEADiff (Qi et al., 2024) demonstrate limited style representation capabilities, often failing to capture intricate style details, especially in the "cup" and "woman" cases. StyleAlign (Hertz et al., 2024) also suffers from content interference, such as the unintended appearance of fruits from the reference image in the "wolf" example. DreamStyler (Ahn et al., 2024) tends to prioritize the prompt text over the reference style. In contrast, our method successfully generates results that both faithfully reproduce the reference style and maintain a strong alignment with the content described in the prompt.

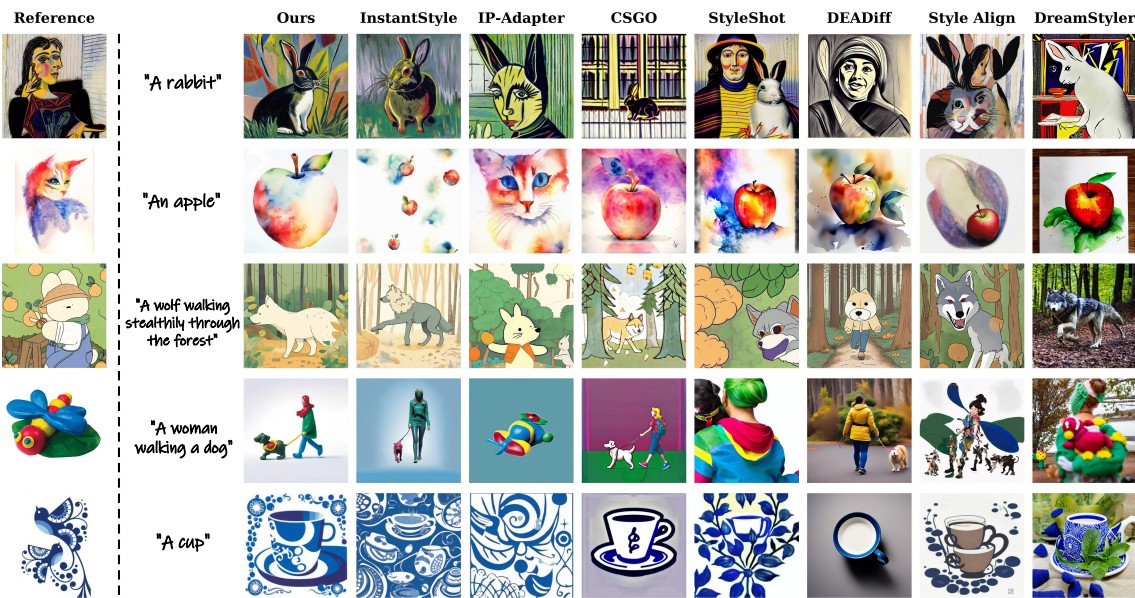

*Figure 5.* **Qualitative comparison with state-of-the-art methods.**

*Table 1.* **Quantitative comparison with state-of-the-art methods.** Our approach achieves the best trade-off between style restoration and text alignment. Bold: best, underline: second best.

| Metric | Ours | CSGO | InstantSt. | StyleShot | IP-Ada. | DEADiff | Style Align | DreamSty. |
|---|---|---|---|---|---|---|---|---|
| †CSD Score ↑ | 53.26 | 47.76 | 50.62 | 52.71 | **78.04** | 39.72 | 50.03 | 20.88 |
| †CLIP-Text ↑ | **28.95** | 26.14 | 28.78 | 25.56 | 19.51 | 27.71 | 26.48 | 28.23 |
| CQ Score ↑ | **10.73** | 9.68 | 10.62 | 9.61 | 8.61 | 10.04 | 9.85 | 9.96 |
| User-study Style % ↑ | **42.97** | 6.12 | 8.73 | 10.77 | 18.02 | 3.29 | 9.41 | 0.68 |
| User-study Text % ↑ | **55.22** | 6.35 | 9.07 | 4.88 | 0.45 | 7.02 | 5.90 | 10.66 |

Note: †CSD Score = 100 × CSD Score, †CLIP-Text = 100 × CLIP-Text.

**Quantitative Comparisons.** Table 1 shows the quantitative comparison with other methods. StyleDistillation achieves the best text prompt alignment performance, as verified by the CLIP-Text score. Additionally, our method outperforms most others in terms of style reproduction capability. In particular, although IP-Adapter performs well in terms of style, it struggles significantly to align with the prompt text. Our method achieves the best trade-off between these two factors, as intuitively demonstrated by the CQ score.

**User Study.** We conduct a user study to evaluate prompt alignment and style similarity from a human perspective. 63 users participated in anonymous voting to select the example they felt best matched the text description and most closely resembled the reference image style, among the outputs generated by different methods. Results are shown in Table 1. More details are provided in the Appendix F.

### 4.3. Discussion

**Ablation Study.** We conduct an ablation study by progressively removing key components from our full model to

*Table 2.* **Ablation by removing key components of our method.**

| Method | CSD Score ↑ | CLIP-Text ↑ |
|---|---|---|
| full model | 53.26 | 28.95 |
| *w/o* PAE | 59.24 | 27.32 |
| *w/o* PAE & $\mathcal{L}_{recon}$ | 58.49 | 27.40 |
| *w/o* PAE & $\mathcal{L}_{recon}$ & $\mathcal{L}_{orth}$ | 61.22 | 27.31 |
| *w/o* PAE & $\mathcal{L}_{recon}$ & $\mathcal{L}_{orth}$ & $\mathcal{L}_{anchor}$ | 33.21 | 26.25 |

assess their individual contributions, with the results presented in Table 2. We first remove the PAE mechanism to focus on the effect of the upstream training losses. The results show that PAE provides a favorable trade-off between style fidelity and prompt alignment, which we further analyze later in this section. Removing the reconstruction loss lowers the CSD score, indicating a loss of style information. This, in turn, shifts the model's focus from style to the text prompt, as reflected by the slight change in the CLIP-Text score. Subsequently, disabling the orthogonality constraint causes overlapping information between the style and content representations. While this counter-intuitively raises the CSD Score, our qualitative results (Figure 7) reveal that

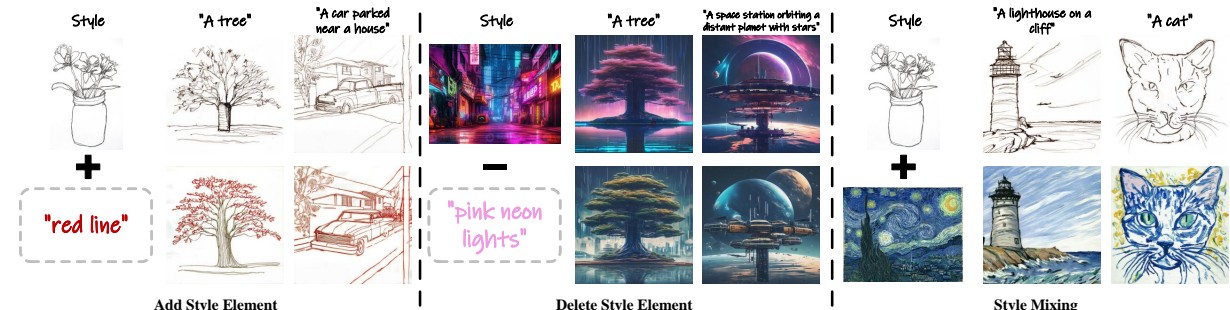

*Figure 6.* **Visual Results of Various Style Manipulation Operations.**

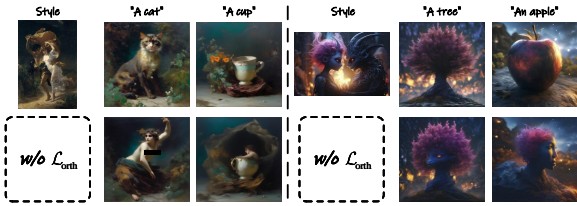

*Figure 7.* **Qualitative ablation for $\mathcal{L}_{\text{orth}}$.** Removing the orthogonality constraint causes severe content leakage from the reference style image into the generated results.

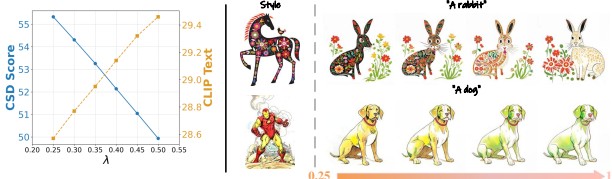

*Figure 8.* **Left**: Effect of varying $\lambda = 0.35$ in PAE. **Right**: Influence of different $\gamma$ on the reconstruction loss.

this is due to stronger interference from the content of the reference image. Finally, without the anchor loss, the model fails to effectively extract style information, leading to a severe degradation in generation quality.

**Analysis of $\gamma$.** In the reconstruction loss defined in Equation 4, the introduction of $\gamma$ plays an important role in stabilizing optimization. Because the anchors used in the anchor loss term are noisy, their loss weights must be set relatively small to avoid overfitting. Consequently, the reconstruction and orthogonality losses contribute more strongly to the optimization gradient. Combined with the tendency of CLIP image embeddings to emphasize content information (Observation 1), this can drive the optimizer toward a degenerate solution where $y_s$ approaches zero and $y_c$ alone explains $f_I^s$. Although this solution roughly satisfies the orthogonality constraint, it discards most style information. By setting $\gamma < 1$, we down-weight the contribution of $y_c$ in the fused representation used by the reconstruction objective, forcing $y_s$ to carry enough stylistic detail to match the reference embedding. As shown in Figure 8, increasing $\gamma$ reduces style fidelity and visible artifacts (e.g., green hues). Quantitatively, the CSD score drops by an average of 13.5%.

**Prompt Alignment Enhancement.** In Figure 8, we study the effect of $\lambda$ when integrating text prompts with style representations. Increasing $\lambda$ strengthens the contribution of the text prompt, which naturally affects the original style representation. As $\lambda$ grows, alignment between the generated images and the text prompts improves, while the style similarity to the reference image gradually decreases. In our experiments, we set $\lambda = 0.35$ to obtain a favorable balance between these two aspects.

### 4.4. Style Manipulation

**Style Edit.** In practical, we often need to adjust the target style in the reference image rather than simply replicate it to meet personalized preferences. As shown in Figure 6, we might want to add more color to the simple lines of a pencil drawing or remove the ubiquitous pink neon lighting from a cyberpunk style. Existing work typically treats style information as a whole, making it difficult to achieve fine-grained control over style. In contrast, our method enables adding or removing of specific style elements using simple text instructions, allowing for precise style editing, as demonstrated in Figure 6. We encode text instructions that describe style elements into the CLIP embedding space and apply a method similar to Equation 6 to adjust style representations. The addition or removal of style elements depends on the positive or negative value and strength of $\lambda$.

**Style Fusion.** Our method also supports fusing the target style with other styles for novel effects. By replacing the text embeddings in Equation 6 with arbitrary style representations, we can incorporate artistic elements from different styles into the target style. As shown in Figure 6, by integrating style cues from Van Gogh's Starry Night, the generated result preserves the line structure while adopting the color tones and brushstrokes characteristic of the painting.

## 5. Conclusion

We presented StyleDistillation, a novel method for single-reference, text-guided style generation. As its core, the StyleDistiller module is built upon two key observations and employs cooperative optimization objectives to derive a

compact style representation while reducing content interference. Along with the PAE mechanism, our framework achieves a strong balance between style fidelity and text alignment, as confirmed by quantitative results and user studies. Moreover, our method enables flexible aesthetic manipulation, such as style fusion, showing its utility for personalized stylization.

## Impact Statement

Our proposed StyleDistillation framework enhances the controllability and flexibility of text-guided stylized image generation, enabling users to manipulate visual styles from a single reference image, not only producing results with high fidelity to the reference style and strong content alignment with the text prompt, but also supporting the creation of personalized styles. This has broad implications for digital art, personalized content creation, and cultural heritage preservation, lowering the barrier to creative visual expression. However, as with many generative models, potential risks include the misuse of stylized content for creating misleading or deceptive imagery, as well as concerns regarding style ownership and copyright when reference images are derived from artistic works. We encourage responsible use and advocate for integrating attribution and provenance mechanisms in future deployments.

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

This appendix provides additional details, analysis, and results to complement the main paper. The content is organized as follows:

- **Section A Explanation of Symbols:** A comprehensive table that explains the mathematical symbols used in the paper and their respective meanings.
- **Section B Method Details:** Thoroughly details our methodology and implementation steps.
- **Section C Additional Details on Our Key Observations:** Extended qualitative comparisons of different style conditioning strategies in Section C.1 and detailed setups for the motivation experiments in Section C.2.
- **Section D Composite Quality Score:** An in-depth discussion of the Composite Quality (CQ) score used to jointly evaluate style fidelity and prompt alignment.
- **Section E Limitations:** Discussion on method limitations.
- **Section F Evaluation Settings and User Study:** Additional information regarding the experiments mentioned in the main article, with specific details on quantitative evaluations and the setup of the user study.
- **Section G Additional Results and Comparisons:** Additional results of our method and qualitative comparisons with other methods.

## A. Explanation of Symbols

To facilitate easy reference, we have compiled a comprehensive list of symbols used in this work in Table 3.

*Table 3.* **Symbol Table for the StyleDistillation**

| Symbol | Meaning |
|---|---|
| $I_s$ | Reference style image |
| $f_I^s$ | CLIP embedding of the reference image $I_s$ |
| $y_s$ | Style representation distilled from the image embedding $f_I^s$ |
| $y_c$ | Auxiliary content representation introduced to facilitate learning a high-quality $y_s$ |
| $d_s$ | Style description generated from the reference image $I_s$ |
| $d_c$ | Content description generated from the reference image $I_s$ |
| $f_t^s$ | CLIP embedding of the style description $d_s$ |
| $f_t^c$ | CLIP embedding of the content description $d_c$ |
| $p$ | Text prompt used during generation |
| $f_t^p$ | CLIP embedding of the text prompt $p$ |
| $y_s^p$ | Combined style-prompt embedding for improved alignment with text prompts |
| $\alpha$ | Strength of content loss in anchor loss |
| $\beta$ | Strength of style loss in anchor loss |
| $\tilde{y}_s$ | L2-normalized version of the style representation $y_s$ |
| $\tilde{y}_c$ | L2-normalized version of the auxiliary content representation $y_c$ |
| $\tilde{y}_{\text{fuse}}$ | Fused representation for reconstruction Loss |
| $\gamma$ | Scalar coefficient controlling the contribution of $y_c$ in the fused representation |
| $\lambda$ | Weight for controlling the strength of the combination of style representation and text prompt |
| $CLIP_i$ | Pre-trained CLIP image encoder |
| $CLIP_t$ | Pre-trained CLIP text encoder |
| $\text{Attn}(\cdot, \cdot)$ | Cross-attention mechanism between text and image |
| $W_s, W_c$ | Linear mappings for style and content representation |
| $Q_t, K_I, V_I$ | Query, key, and value matrices in the cross-attention mechanism |
| $d_k$ | Dimension of the key tensor in the cross-attention mechanism |
| $\epsilon_\theta(z_t, t, CLIP_t(d_c), y_s)$ | Predicted noise by the UNet model |
| $\mathcal{L}_{\text{denoise}}$ | Denoise loss |
| $\mathcal{L}_{\text{orth}}$ | Orthogonality loss |
| $\mathcal{L}_{\text{style}}$ | Style loss based on observation 1 |
| $\mathcal{L}_{\text{content}}$ | Content loss based on observation 2 |
| $\mathcal{L}_{\text{anchor}}$ | Anchor loss constructed by style loss and content loss |
| $\mathcal{L}_{\text{recon}}$ | Reconstruction loss |
| $\mathcal{L}_{\text{Distillation}}$ | Final loss function for training the StyleDistiller module |
| $x_{\text{sty}}^i$ | CSD score for the $i$-th method (measuring style similarity) |
| $x_{\text{text}}^i$ | CLIP-Text score for the $i$-th method (measuring content alignment with text prompts) |
| $\sigma_{\text{sty}}$ | Standard deviation of the CSD scores |
| $\sigma_{\text{text}}$ | Standard deviation of the CLIP-Text scores |
| $CQ_i$ | Composite Quality (CQ) score of the $i$-th method, combining both style and text alignment evaluation |

## B. Method Details

Given a reference style image, our method first employs a trainable StyleDistiller module to extract a fine-grained style representation. By further introducing Prompt Alignment Enhancement in the inference stage, we are able to generate images that not only faithfully preserve the reference style but also exhibit strong alignment with the input text prompt. The detailed procedures for the training and inference stages are summarized in Algorithm 1 and 2, respectively.

---

**Algorithm 1** Training StyleDistiller

---

**Input:** reference style image $I_s$, text encoder $CLIP_t$, image encoder $CLIP_i$, multimodal large language model MLLM, pretrained Stable Diffusion with IP-Adapter $\epsilon_\theta$, noise latent $z_t$ corresponding to $I_s$ after adding noise $\epsilon$ at timestep $t$, Trainable StyleDistiller $g(\cdot)$
**Output:** style representation $y_s$ corresponding to $I_s$
  1: Freeze the weights of $\epsilon_\theta, CLIP_t, CLIP_i$
  2: Encode reference image embedding $f_I^s \leftarrow CLIP_i(I_s)$
  3: Generate content and style descriptions $d_c, d_s \leftarrow$ MLLM.describe$(I_s)$
  4: Encode textual embeddings $f_t^c \leftarrow CLIP_t(d_c), \quad f_t^s \leftarrow CLIP_t(d_s)$
  5: **while** training not converged **do**
  6:      Predict style and content representations $y_s, y_c \leftarrow g(f_I^s, f_t^s, f_t^c)$
  7:      Compute orthogonality loss $\mathcal{L}_{\text{orth}} = \mathbb{E}\left[\text{ClipSim}(y_s, y_c)\right].$
  8:      Compute anchor loss $\mathcal{L}_{\text{anchor}}$ by Equation (3)
  9:      Compute reconstruction loss $\mathcal{L}_{\text{recon}}$ by Equation (4)
10:      Predict noise $\hat{\epsilon} \leftarrow \epsilon_\theta(z_t, t, f_t^c, y_s)$
11:      Compute denoising loss $\mathcal{L}_{\text{denoise}} = \|\hat{\epsilon} - \epsilon\|_2^2$
12:      Aggregate total loss $\mathcal{L}_{\text{Distillation}} = \mathcal{L}_{\text{denoise}} + \mathcal{L}_{\text{orth}} + \mathcal{L}_{\text{anchor}} + \mathcal{L}_{\text{recon}}$
13:      Update parameters of $g(\cdot)$ using gradient of $\mathcal{L}_{\text{Distillation}}$
14: **end while**
15: **Return** $y_s$

---

**Algorithm 2** Inference with Prompt Alignment Enhancement

---

**Input:** style representation $y_s$ corresponding to $I_s$, text prompt $p$, pretrained Stable Diffusion with IP-Adapter $\epsilon_\theta$
**Output:** Stylized image $\hat{I}$ that reflects the style of $I_s$ while remaining faithful to the content of $p$
  1: Encode user text prompt $f_t^p \leftarrow CLIP_t(p)$
  2: Compose style-prompt embedding through Prompt Alignment Enhancement: $y_s^p \leftarrow y_s + \lambda \cdot f_t^p$
  3: Generate stylized image $\hat{I} \leftarrow \epsilon_\theta.\text{generate}(p, \text{style} = y_s^p)$
  4: **Return** $\hat{I}$

---

It is worth noting that, in the implementation of StyleDistiller, we encode the reference style image $I_s$ with the CLIP image encoder and use both its last hidden states and its final global embedding. Since the former preserves richer token-level visual information, it is used in the cross-attention computation, enabling the model to capture fine-grained visual details from the reference image when extracting the style representation. The latter lies in the shared CLIP text-image embedding space, and is therefore used for computing the optimization losses. In our OpenCLIP ViT-bigG-14 implementation, the text features $f_t^s$ and $f_t^c$ are vectors with shape $[1, 1280]$. The last hidden states of the reference image form a matrix with shape $[257, 1664]$, which is mapped by a learnable linear projection layer to $[257, 1280]$ before cross-attention, where it serves as the key and value. The text features serve as the query in this cross-attention computation. The cross-attention output is an intermediate feature with shape $[1, 1280]$, which is then passed through a linear layer to obtain the final style representation $y_s$ and auxiliary content representation $y_c$, both with shape $[1, 1280]$. The global CLIP image embedding used for loss computation is also a vector with shape $[1, 1280]$. For clarity and readability, the main paper denotes these image features encoded from the same reference image uniformly as $f_I^s$, abstracting away this implementation-level distinction.

## C. Additional Details on Our Key Observations

### C.1. Comparison of Style Conditioning Strategies

As discussed in the introduction of the main paper, due to the coupling of style and content information in the reference image, directly using it as a style constraint causes style-irrelevant content information to severely interfere with the generated

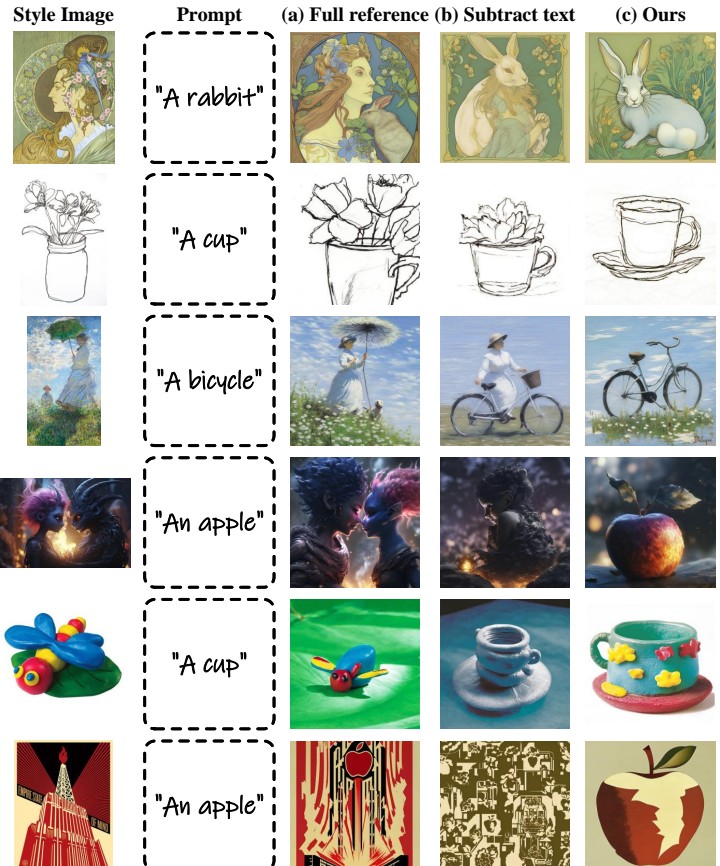

*Figure 9.* **Qualitative comparison of style conditioning strategies.**

results. Simply subtracting the content description from the reference image embedding not only fails to effectively eliminate content interference, but also tends to damage the style information within the reference image. To visually demonstrate these limitations, we conduct a comparative experiment, with a subset of the results presented in the main paper. Specifically, we use a pre-trained CLIP image encoder to obtain reference image embeddings. From these, we derive three types of style conditions: the original reference image embeddings, the embeddings after subtracting the corresponding content description text embeddings, and the style representations produced by our method. Each of these conditions is then used as a style constraint to guide image generation together with the same text prompt.

Additional results are shown in Figure 9. As illustrated in column (a), when the reference image is directly used as the style constraint, style-irrelevant content information from the reference consistently appears in every generated image, resulting in complete misalignment with the text prompt. In the column (b), where the content text description is subtracted, this operation still fails to effectively remove content interference (e.g., the inappropriate appearance of hair, flowers, and characters in the first four cases), and at the same time severely degrades the reference style details (e.g., the last three cases exhibit varying degrees of loss in the reference style's color and lighting). These shortcomings further hinder alignment with the text prompt, as illustrated by the disrupted composition in the last case. These issues motivate us to further explore the CLIP embedding space from a style perspective. The resulting StyleDistillation approach demonstrates clear advantages in generating images that simultaneously exhibit high style fidelity and strong alignment with the prompts, as shown in column (c).

### C.2. Further Details on the Motivation Experiments

In the motivation section of the main paper, we conduct a series of experiments and obtain two key observations that form the basis of our method: (1) the effect of style on image embeddings exhibits cross-content consistency in the CLIP space, and (2) text descriptions cannot be perfectly aligned with all visual elements in the reference images.

**Reference Image** 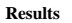 **Results**

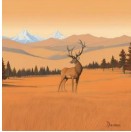 **Content description:** A deer with prominent antlers stands on orange grass, surrounded by tall trees under a blue sky with distant mountains.

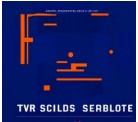 **Style description:** The image's visual style features bold, flat colors with a modern, graphic-cartoon aesthetic. It employs a restricted, clean color palette dominated by deep blues and vibrant oranges, contrasted for high visual impact. There is prominent use of geometric shapes and smooth gradients to convey depth and form. Sharp, clean edges suggest digital precision, while subtle texturing adds a tactile quality. The lighting is diffuse, emphasizing shape over detail, and the composition is balanced with a strong sense of symmetry and rhythm.

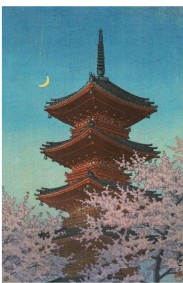 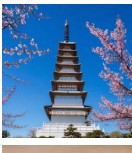 **Content description:** A multi-tiered pagoda stands against a blue sky with a crescent moon, surrounded by blooming cherry blossom trees.

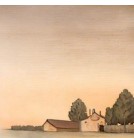 **Style description:** The image's visual style is characterized by delicate, precise linework and a harmonious composition. The color palette combines soft pastels and warm, earthy hues, creating a tranquil atmosphere. Subtle gradients in the sky suggest depth. The texture resembles fine printmaking with small dot patterns. Lighting is gentle, providing a serene illumination, while the intricate detailing emphasizes structure. The overall tone is calm and contemplative, reflecting traditional techniques reminiscent of ukiyo-e.

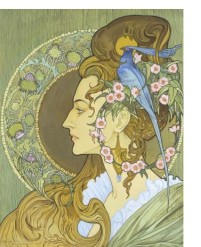 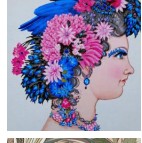 **Content description:** A profile of a woman, her intricate hair adorned with a blue bird and multiple pink flowers. She wears detailed earrings.

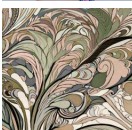 **Style description:** The image's visual style features intricate linework and flowing curves characteristic of Art Nouveau. The color palette includes muted earth tones with soft hues of green, pink, and blue. Textures are ornate and decorative, with a delicate, almost tapestry-like quality. The composition emphasizes fluidity and movement through organic shapes and stylized patterns. Brush strokes are smooth, contributing to an overall elegant and cohesive appearance. The use of light is subtle, focusing more on pattern and line differentiation than on depth or shadow.

*Figure 10.* **More Details on Observation 2 Experiment:** Including the example reported in the main paper, we show several reference style images; for each reference, we provide the content and style descriptions used in our experiment and the corresponding images generated from each description.

For the first observation, we use UnlearnCanvas (Zhang et al., 2024b), a stylization dataset consisting of 24,000 images. It is constructed by transferring 400 content images (20 different categories, each containing 20 images) into 60 different styles. This construction naturally allows us to partition the samples from two perspectives: by style, which yields 60 style groups, and by content, which yields 400 content groups. For the original image embeddings in the dataset, we compute the overall silhouette score (Rousseeuw, 1987) under these two partitions. The silhouette score is a widely used clustering quality metric that jointly considers the compactness within each cluster and the separation between neighboring clusters. For a sample $i$, let $a(i)$ denote the mean distance from $i$ to all other samples in the same cluster, and let $b(i)$ denote the mean distance from $i$ to samples in the nearest different cluster. The silhouette score of $i$ is defined as $s(i) = \frac{b(i)-a(i)}{\max\{a(i),b(i)\}}$. The overall silhouette score is then obtained by averaging $s(i)$ over all samples. We treat each style/content group as a cluster and compute silhouette scores in the CLIP embedding space using Euclidean distance. As reported in the main paper, the mean silhouette score is 0.137 when grouping by content and only 0.035 when grouping by style, indicating that content information plays the primary role in shaping the embeddings.

We further analyze the nearest-neighbor structure of the embeddings. For each sample, we find its nearest neighbor and measure the fraction of pairs that share the same label. When using content labels, this fraction is 0.726, whereas it drops to 0.271 when using style labels, which reinforces the above conclusion. To mitigate interference from content information and better observe style, we apply a de-projection step to the embeddings. For each content group, we first compute the mean embedding over all images of that content and regard it as the content direction. We then remove from each sample its projection on this direction. This procedure is conceptually similar to the way IP-Composer (Dorfman et al., 2025) extracts visual concepts. Although it does not perform fine-grained style extraction, it is sufficient for analyzing style in the CLIP space. The t-SNE visualizations in the main paper show that samples sharing the same style form much tighter clusters

across different contents, revealing the cross-content consistency of style. Quantitatively, the mean silhouette score under the style-based partition increases to 0.102, and the fraction of samples whose nearest neighbor has the same style label rises to 0.935.

For the second observation, we use the same set of reference style images as in the main experiments of the main paper, as illustrated in Figure 14. Following the same procedure and instructions described in Section F, we generate for each reference image a content description and a style description. To investigate whether these textual descriptions can serve as direct substitutes for the visual information in the images, we encode the content and style descriptions with a pretrained CLIP text encoder and use the resulting embeddings to guide image generation. As shown in the main paper, we consistently observe that generations conditioned on content descriptions better reproduce the corresponding content information of the reference images. However, they still fail to reconstruct all visual details of the references, such as background elements and local structures. We present additional examples for more reference images in Figure 10, together with their content and style descriptions, to further support this observation.

## D. Composite Quality Score

In our evaluation, we aim to measure both the style fidelity of the generated images and their content alignment with the given text prompts. Concretely, we employ two complementary metrics: the CSD score for assessing style fidelity, and the CLIP-Text score for measuring consistency with the prompt. An ideal method should perform well on both metrics. However, in practice, we observe a trade-off: improving style fidelity often leads to diminished prompt alignment, and vice versa. A plausible explanation is that diffusion models condition generation on both style and text prompt, increasing the emphasis on one inherently compromises the other.

Therefore, it is crucial not only to achieve strong individual performance on both axes but also to maintain a favorable balance between them. However, since these metrics are inherently independent, directly evaluating this trade-off is nontrivial. A naïve approach is to visualize methods in a 2D coordinate space with CSD score and CLIP-Text score as the axes, where proximity to the top-right corner indicates superior performance. This is similar to calculating the Euclidean distance between different coordinate points and the origin, as shown in Figure 11. However, it is not difficult to see that due to the difference in the scale of the two metrics, this direct calculation is not fair.

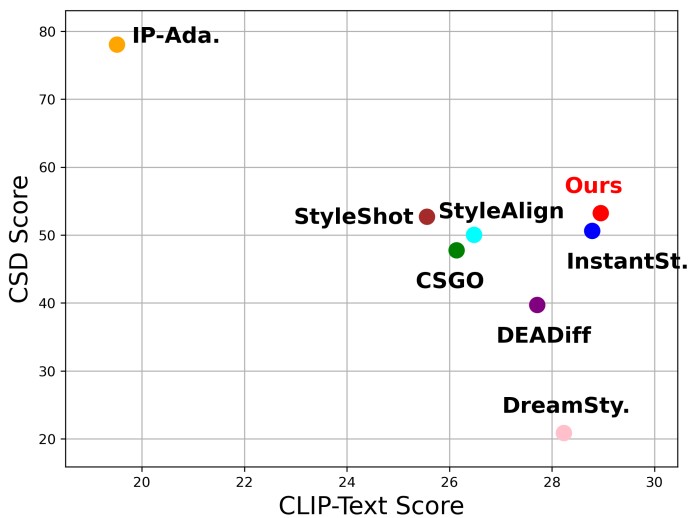

*Figure 11.* **Performance of CSD and CLIP-Text scores in Text-guided stylized image generation.**

To address this, we propose Composite Quality (CQ) Score that accounts for this imbalance. Specifically, we first compute the standard deviation of each metric across all evaluated methods, and normalize individual scores by their respective standard deviations. This normalization—akin to Z-score transformation—ensures comparability by mapping scores to distributions with unit variance. We then calculate the Euclidean distance from the origin in this normalized space. This distance, which we term the Composite Quality Score, serves as a unified indicator of overall performance. Intuitively, CQ favors methods that improve both style fidelity and prompt alignment, rather than over-optimizing a single metric while neglecting the other. Given a set of evaluation results from different methods, the Composite Quality (CQ) score of the $i$-th method can then be calculated as:

$$\mathrm{CQ}_i = \sqrt{\left(\frac{x^i_{\mathrm{sty}}}{\sigma_{\mathrm{sty}}}\right)^2 + \left(\frac{x^i_{\mathrm{text}}}{\sigma_{\mathrm{text}}}\right)^2},$$

where $x^i_{\mathrm{sty}}$ and $x^i_{\mathrm{text}}$ represent the CSD and CLIP-Text scores of the $i$-th method, and $\sigma_{\mathrm{sty}}$, $\sigma_{\mathrm{text}}$ are the corresponding standard deviations computed over all evaluated methods.

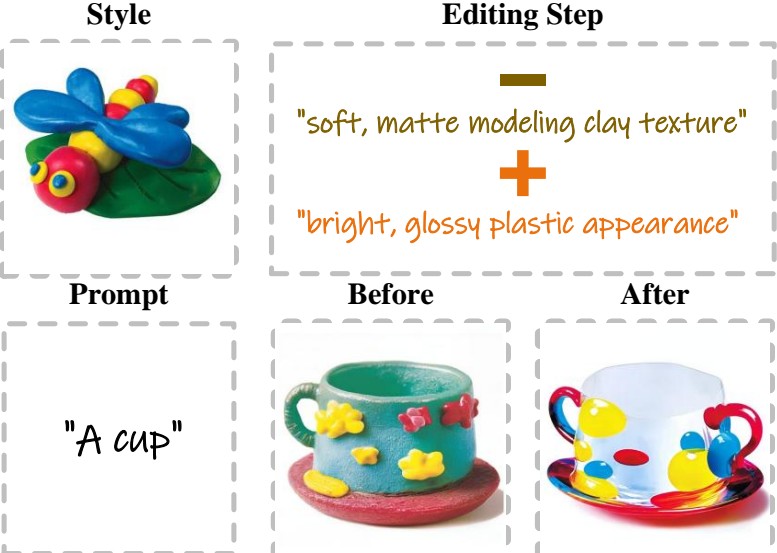

*Figure 12.* **Failure case of style editing.** We attempt to remove the "soft, matte modeling clay texture" of the reference style and replace it with a "bright, glossy plastic appearance". While the material becomes glossier, the edit overshoots the intended change, introduces exaggerated color/highlight shifts, and slightly disrupts the original composition.

## E. Limitations

Our method performs strongly on text-guided stylized image generation and naturally supports a range of personalized aesthetic applications, including style editing and style fusion. In style editing scenarios, users can flexibly adjust attributes such as color, material, or lighting of the reference style using short textual instructions. However, the current style editing operation still has certain limitations. For most common editing instructions, our method reliably produces the intended modifications, but in a small number of particularly challenging cases the generated images still fall short, as illustrated by the failure example in Figure 12. Such failures typically occur when fine-grained control over multiple style attributes is required while the textual instructions remain relatively ambiguous, causing the edits to overshoot the desired change and slightly disturb the original composition. We observe that adjusting the editing strength or rewriting and decomposing the instructions often alleviates this issue, suggesting that the limitation is more related to the current editing granularity than to any inherent inability of our method to represent these style variations. Importantly, style editing in text-guided stylized image generation remains largely underexplored, and we believe our work provides a solid foundation for future advances in this direction. In ongoing work, we plan to explore more expressive or multimodal control strategies to improve the fidelity and granularity of style editing.

## F. Evaluation Settings and User Study

As described in the experimental setup of the main paper, our quantitative evaluation was conducted on a test set comprising 40 reference style images and 30 text prompts, spanning a range of difficulty levels from simple to complex. The full set of reference styles is illustrated in Figure 14, and the complete list of text prompts is provided in Figure 13. For each method, we generated a total of 6,000 images to ensure a comprehensive comparison. In addition, our approach leverages GPT-4o to generate content and style descriptions for each reference image. The MLLM instruction we used for this purpose is shown in Figure 15.

To complement the quantitative analysis, we conducted a user study to gather subjective evaluations of the different methods. Participants were presented with 14 image-text pairs, each consisting of a reference style image and a corresponding text prompt. For each pair, participants were asked to independently select the method that best matched the reference style and the method that best aligned with the prompt text, respectively. This dual-question format allowed for a more fine-grained assessment across the two core objectives of stylized image generation. To ensure fair and consistent evaluations, all participants received clear task instructions prior to the study. In total, 63 volunteers from diverse backgrounds participated, resulting in 1,764 valid responses. The questionnaire design is showed in Figure 16.

| A tree | A rabbit | A dog |
|---|---|---|
| A cat | An apple | A bicycle |
| A bench | A table | A cup |
| A robot | A sailboat docked in a harbor | A car parked near a house |
| A lighthouse on a cliff | Ancient temple hidden in jungle | Flower field in spring |
| A bus stopping at a station | A bird flying in the sky | A bridge connecting two hills |
| A woman walking a dog | A child playing with a ball | A wolf walking stealthily through the forest |
| A dragon soaring through a cloudy sky at sunset | A lonely cabin covered in snow in a dense forest | A student walking to school with backpack |
| A person jogging along a scenic trail | A medieval castle standing tall against a stormy sky | A serene sunset over a calm lake |
| A wild west town with dusty streets and wooden saloons | A space station orbiting a distant planet with stars | A majestic lion resting in the golden savannah under a tree |

*Figure 13.* **The complete list of text prompts.**

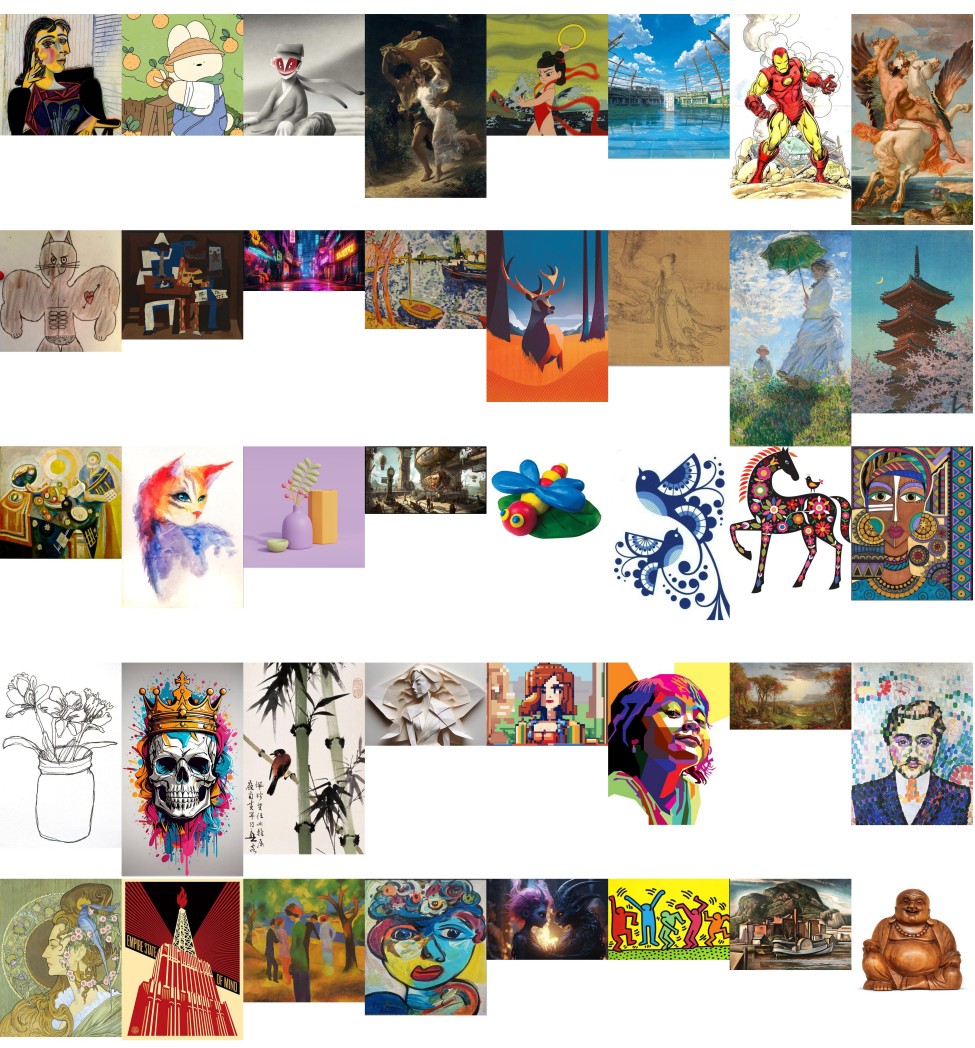

*Figure 14.* **The full set of reference styles.**

Style description：I will provide you with a single image that represents a distinct visual style. Your task is to describe the visual style as thoroughly and specifically as possible. Focus only on stylistic elements such as brush strokes, texture, color palette, lighting, composition, visual tone, and artistic techniques. Do not describe any objects, characters, scenes, or layouts. Avoid narrative interpretation or emotional inference that is not visually grounded. Concentrate purely on how it looks, not what it depicts. Your description should be concise, under 77 tokens, and must not reveal whether the description is based on one image or multiple images.

Content description：I would like you to describe the picture in 15–30 words. The description should focus only on what is shown in the picture (e.g., the scene, objects, and layout), not on the style of the picture (e.g., brush strokes, texture, etc.). Your description must be limited to the picture's content and avoid any interpretation (e.g., a picture of a cat should be described as "a black-and-white cat lying on the grass," not "a cat on the grass, which may suggest a sad atmosphere"). Do not include any information about the picture's style. In other words, the description should not suggest how the picture looks stylistically. Instead, it should offer a clear, detailed account of the content — objects, layout, and scene — so that someone reading it can form an accurate idea of what is in the image.

*Figure 15.* **The MLLM instruction used to generate style and content description.**

(a).Choose the image that best matches the **text** description from the 8 images below.

(b).Choose the image that best matches the **style** from the 8 images below.

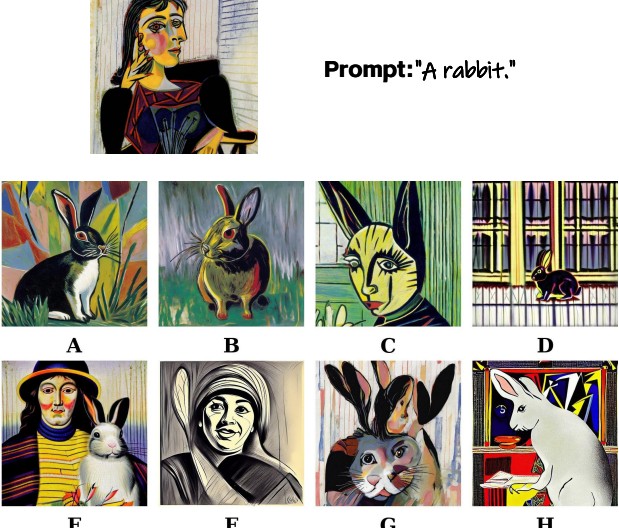

*Figure 16.* **Questionnaire format used in the user study. Each option corresponds to the output of a specific method conditioned on the same reference style image and text prompt.**

# G. Additional Results and Comparisons

**Additional Comparisons**    Figure 17 and 18 present a comprehensive qualitative comparison between our method and several state-of-the-art approaches, evaluated across a wide variety of reference styles and text prompts with varying complexity. As illustrated, StyleDistillation consistently produces high-quality images that not only exhibit a faithful reconstruction of the intricate visual characteristics of the reference style—such as color palette, texture, brushwork, and lighting—but also adhere closely to the semantic content specified in the prompt. This consistent performance across diverse styles and prompts underscores the robustness of our approach. In contrast, existing methods often struggle with either overfitting to the reference style at the expense of prompt alignment, or vice versa, failing to preserve the distinctive aesthetic of the reference. Notably, StyleDistillation excels in both dimensions simultaneously and, more importantly, achieves a well-calibrated balance between them—a critical yet often overlooked aspect in stylized image generation. This balance is particularly valuable in practical applications, where both stylistic fidelity and semantic relevance are essential.

**Additional Results**    Figures 19 and 20 present additional visual results generated by our method using text prompts of varying difficulty across a wide range of reference styles. Rather than merely copying the reference style, our approach adaptively refines stylistic details based on the semantic content of the prompt, while faithfully preserving the core characteristics of the reference style to suit diverse contexts. This flexibility enables more nuanced and context-aware generations, rather than static style transfer. The results clearly underscore the robustness of our approach, as it consistently produces images that achieve high-fidelity style reproduction and strong alignment with text prompt across a broad and challenging spectrum of style categories. Furthermore, Figures 21 and 22 provide more examples of our style editing and style fusion capabilities, respectively. These results highlight the versatility of our framework in supporting fine-grained style manipulation, including the addition, removal, or blending of specific style attributes, all driven by intuitive inputs such as text or multiple style references. Collectively, these qualitative results demonstrate not only the aesthetic quality and accuracy of our method, but also its practical value in enabling a wide range of user-customizable and creative applications in stylized image generation.

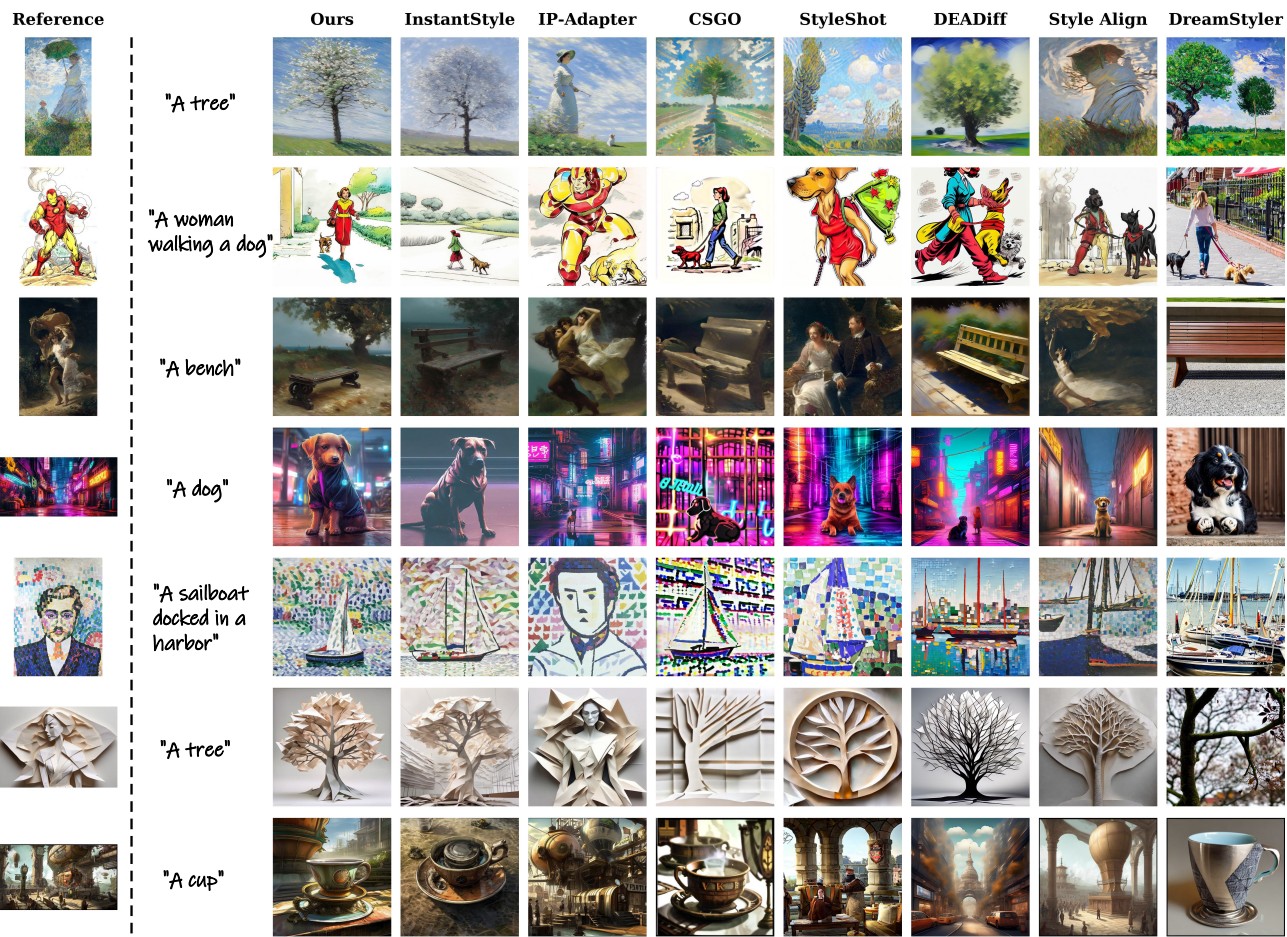

*Figure 17.* **Additional qualitative comparison with other state-of-the-art methods.**

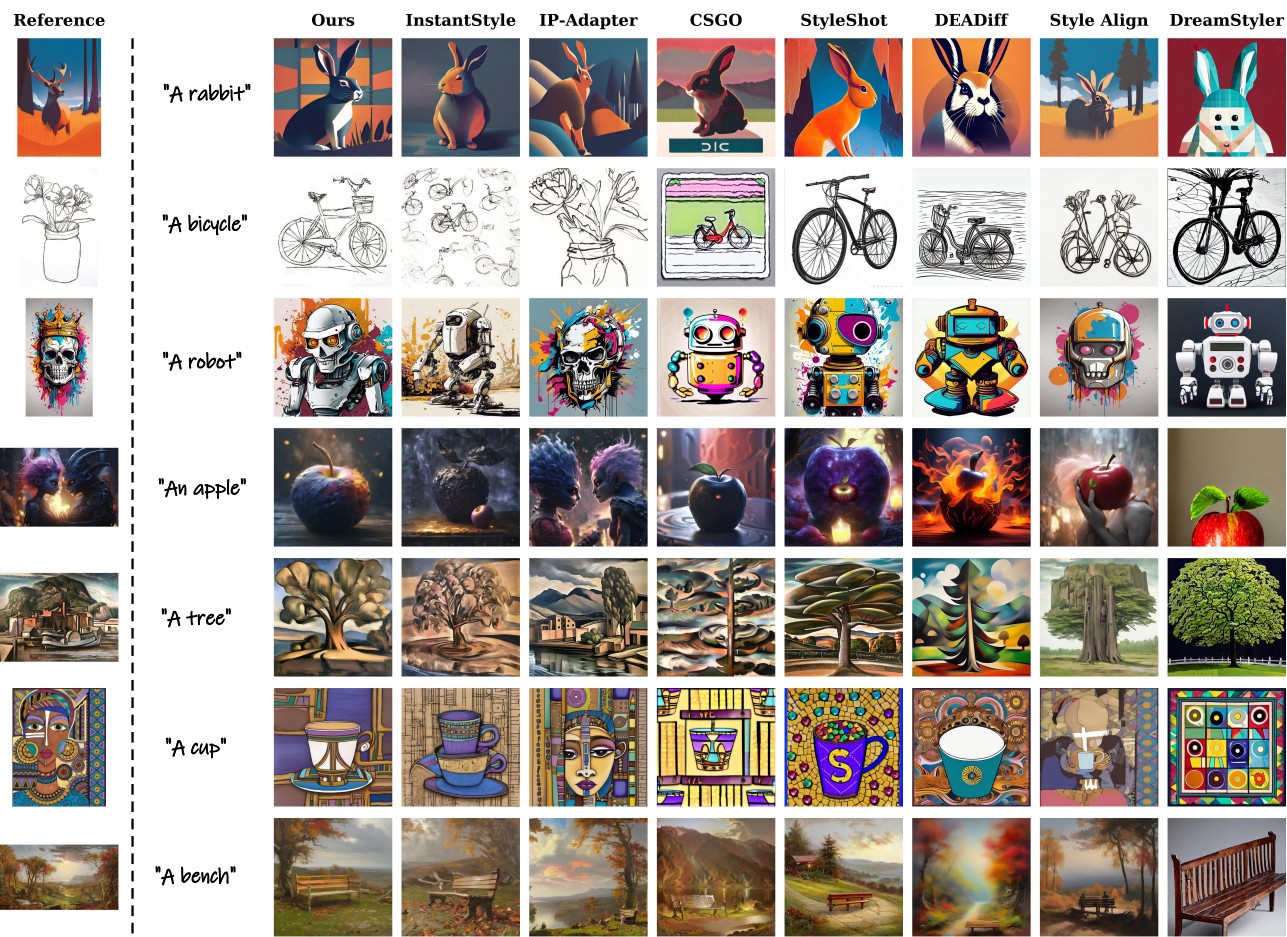

*Figure 18.* **Additional qualitative comparison with other state-of-the-art methods.**

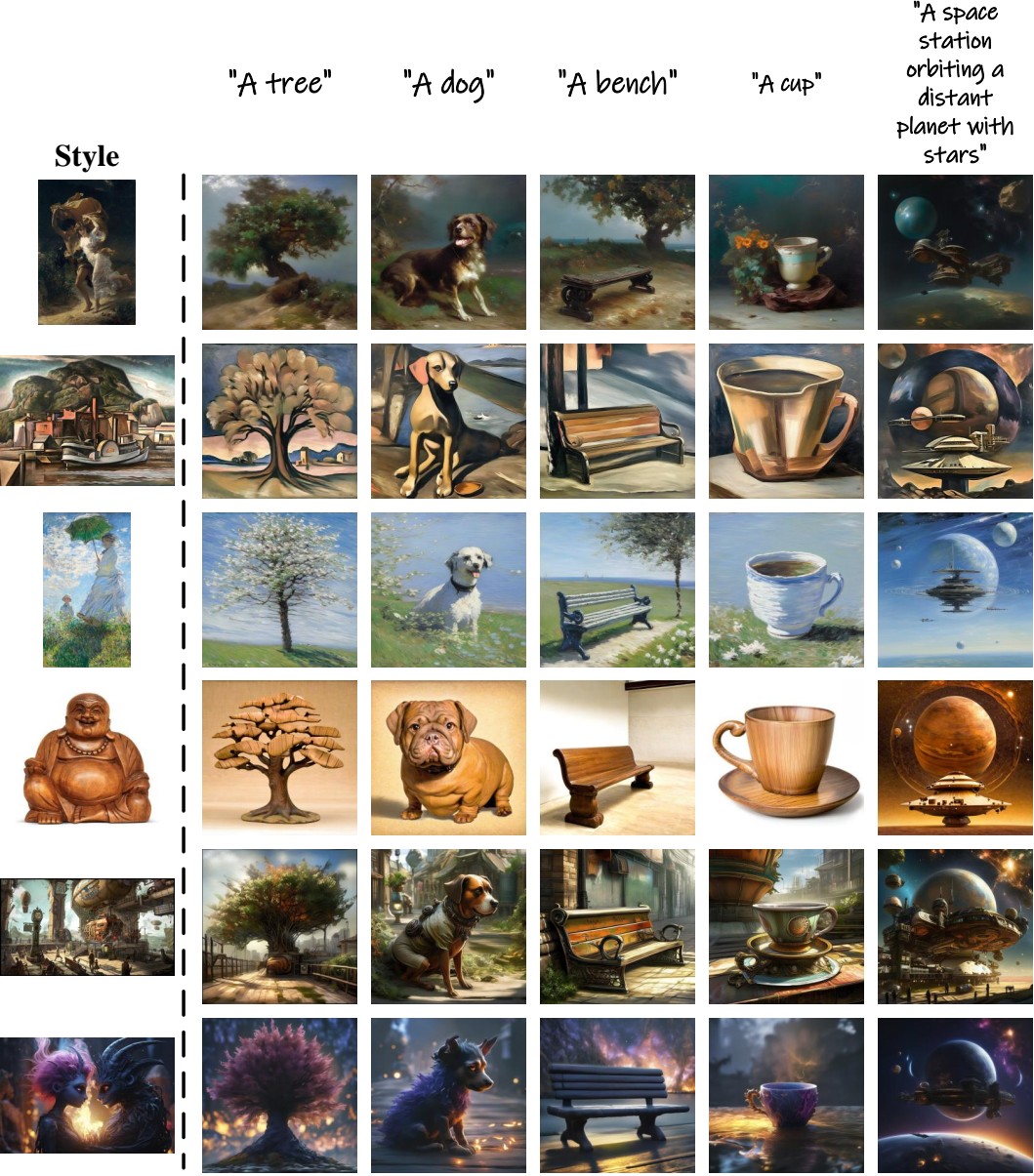

*Figure 19.* **Additional results of our method.**

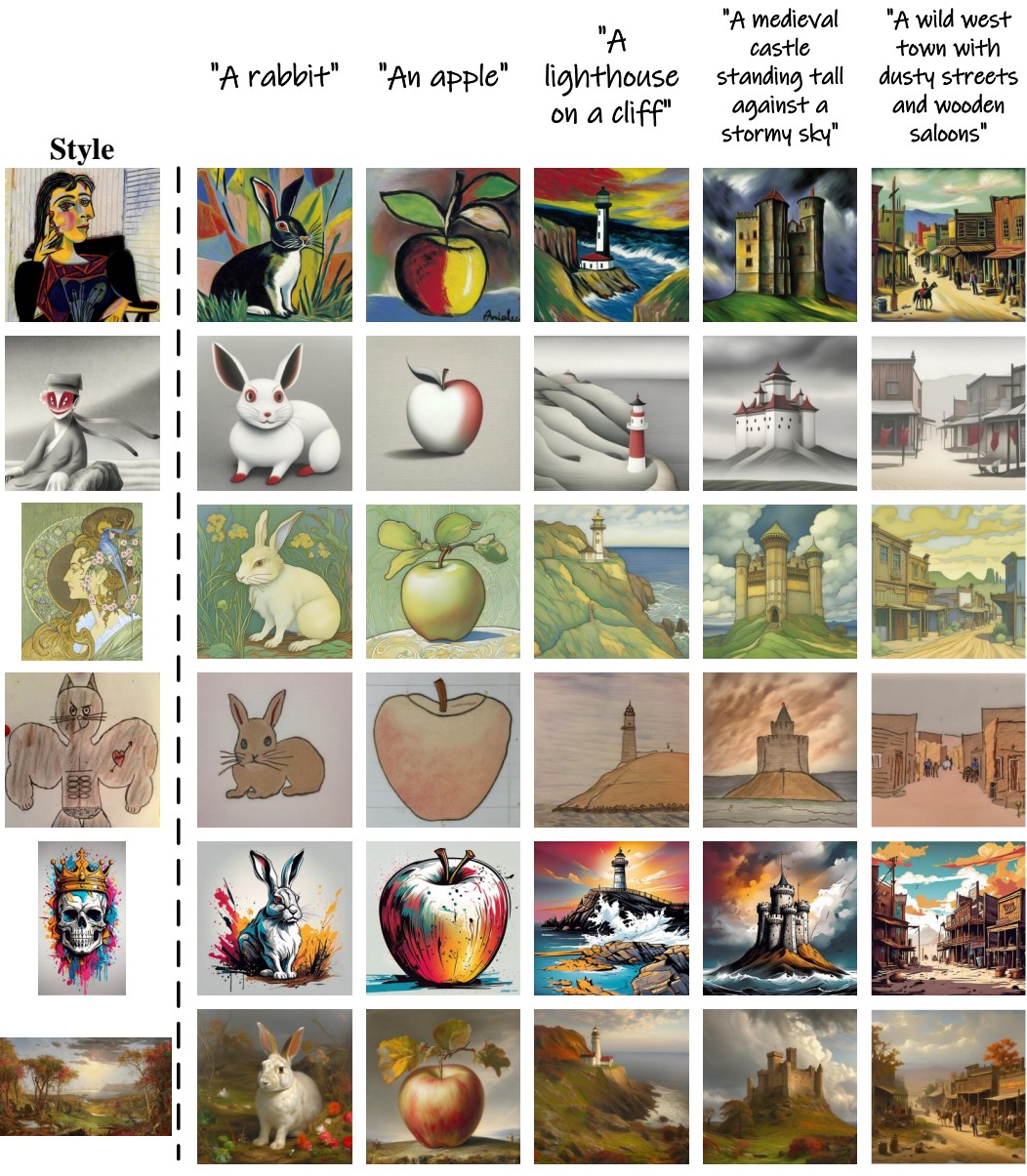

*Figure 20.* **Additional results of our method.**

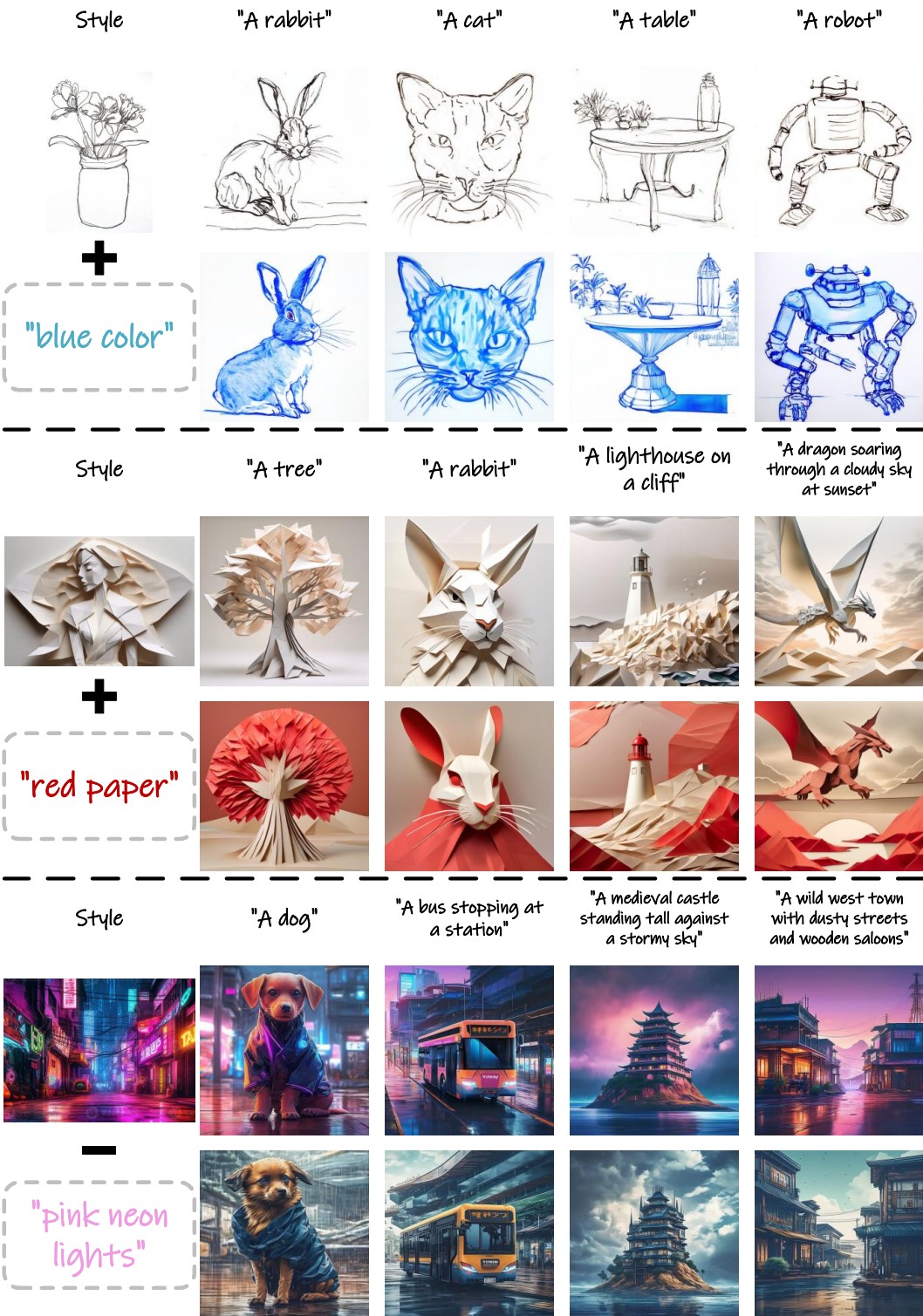

*Figure 21.* **Additional results of style editing.**

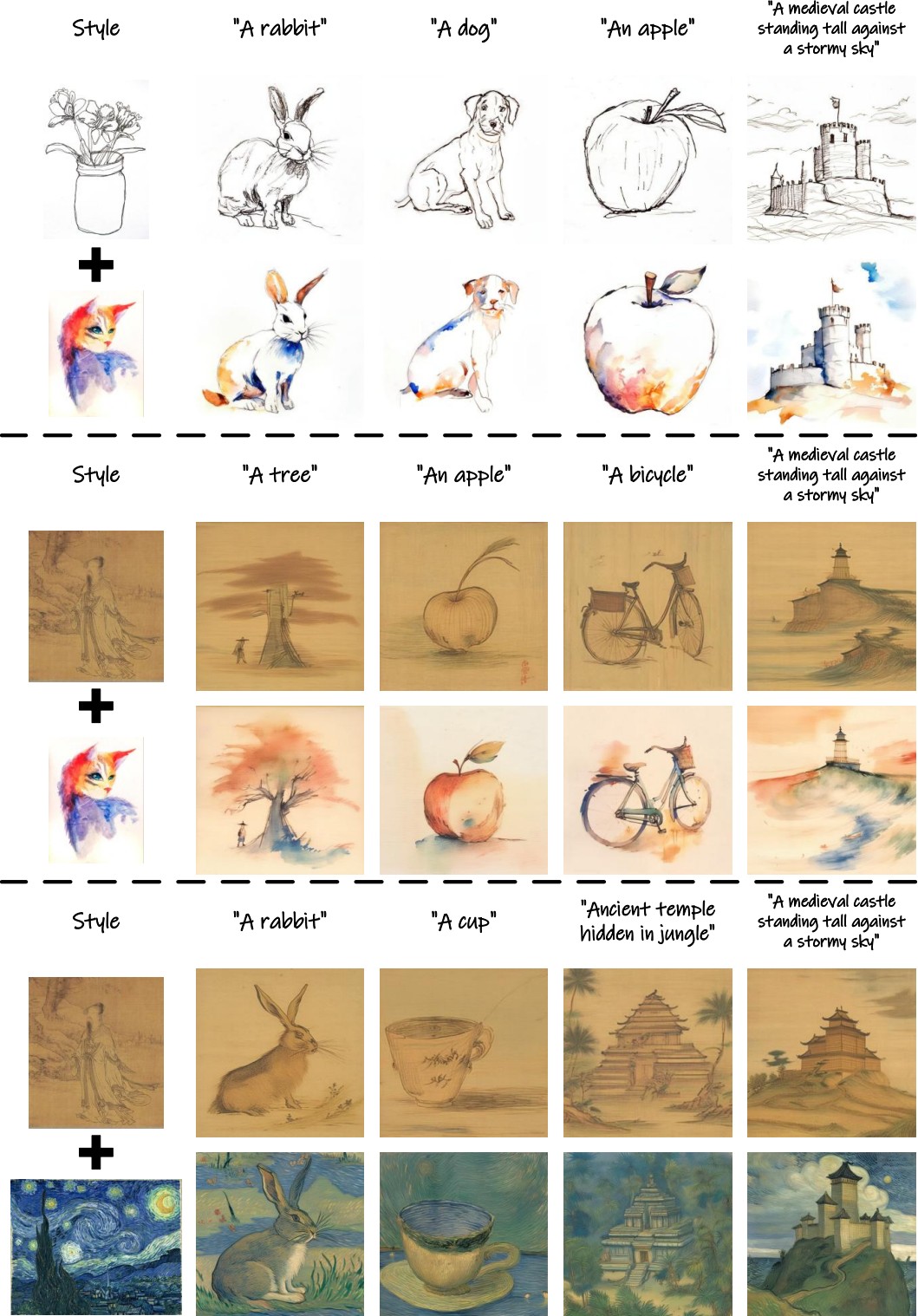

*Figure 22.* **Additional results of style fusion.**

