# OpenReview forum: "StyleDistillation: A New Insight of Image Style Enables Personalized Aesthetic Manipulation"
_ICML.cc/2026/Conference — ICML 2026 regular_

### Official Review · Reviewer_hXvm · 2026-02-14

**Soundness:** 2
**Presentation:** 3
**Significance:** 3
**Originality:** 2
**Overall Recommendation:** 4
**Confidence:** 3

**Summary:**

This paper introduces Style Distillation, a framework designed to disentangle style and content from a single reference image for text-guided stylization. Based on observations of the CLIP embedding space, the authors propose a targeted optimization strategy.
Key contributions include:
- StyleDistiller Module: Features an auxiliary content branch to explicitly decouple content features during training.
- Loss Composition: Anchor, Orthogonality, and Reconstruction losses grounded in geometric and semantic priors to extract a purified style representation $y_s$.
- Prompt Alignment Enhancement (PAE): A mechanism using a factor $\lambda$ to balance style fidelity and prompt alignment during inference.
The framework also enables personalized applications such as style editing and fusion.

**Compliance With Llm Reviewing Policy:**

Affirmed.

**Final Justification:**

he authors' rebuttal has largely addressed my concerns, and I will raise my score accordingly.

**Key Questions For Authors:**

- Given the current UNet-based implementation, can this framework be generalized to DiT (Diffusion Transformer) architectures?
- While Figure 7 shows the effect of the orthogonality loss, such constraints can lead to instability or collapse. How do you ensure a stable decoupling within only 300 training steps?
- In some cases in Figure 6 (e.g., the "Cat" example), the structural changes from the input are significant. How do the authors define the boundary between "style" and "structure" in their extraction process?

**Limitations:**

yes

**Strengths And Weaknesses:**

Strengths:
- The motivation is built on quantitative observations. Silhouette score analysis proves content's dominance in CLIP space while style acts as a consistent offset, providing a solid theoretical foundation.
- The paper is exceptionally well-written and organized. Diagrams (e.g., Figure 3 and 11) effectively illustrate the workflow and metric logic.
- The decoupling approach is intuitive and reasonable, producing high-quality visual results with strong style fidelity.

Weaknesses:
- The work is significantly inspired by prior art like InstantStyle. Despite the orthogonality loss, the framework represents an incremental improvement over existing Adapter-based architectures.
- The method heavily relies on MLLM-generated (e.g., GPT-4o) descriptions. Inaccuracies in these descriptions can directly mislead the $\mathcal{L}_{content}$ objective. Additionally, the performance of unified models (like "Nano Banana") in this domain remains unexplored.
Hyperparameters $\lambda$ and $\gamma$ require precise tuning. Figure 8 shows that style fidelity (CSD) fluctuates sharply when parameters deviate, suggesting potential generalizability issues across diverse styles.

---

> ### Author Rebuttal · Authors · 2026-03-31
>
> We thank the reviewer for the feedback. Supplementary tables and qualitative results:
>
> https://anonymous.4open.science/r/StyleDistillation-D4AE/hXvm.md
>
> # Q1 Prior work
> Indeed, our work and InstantStyle study the same task: style-content disentanglement in text-guided stylization. However, they differ fundamentally in formulation:
>
> |Aspect|InstantStyle|Ours|
> |-|-|-|
> |Motivation|Assumes text sufficiently captures image content | Two observations: (1) cross-content style consistency in CLIP space (2) text-image misalignment|
> |Style extraction|Subtracts content text, leading to style distortion and incomplete content removal (Fig. 2)| Treats text as soft guidance and distills style through jointly constrained objectives|
> |Style injection|Injected into specific SDXL layer|Balanced with prompt in CLIP space|
> |Application|Style preservation only|Style preservation, editing, and fusion|
>
> Thus, our contribution is not primarily a different adapter architecture, but an observation-driven style disentanglement formulation: explicit style/content factorization, jointly constrained objectives, and an inference-time style-prompt balancing mechanism.
>
> # Q2 MLLM, unified models, and hyperparameters
> Our method does not critically depend on MLLMs. Text serves only as soft semantic guidance: global CLIP features of the content description provide overall semantics, while the final style representation is jointly determined by multiple objectives; thus mild ambiguity has limited impact. Replacing ChatGPT-4o with a locally deployed Qwen2.5-7B causes no significant performance change (see anonymous link), indicating that MLLMs are a replaceable prior source rather than a critical dependency. Users can also provide descriptions directly.
>
> For unified models, we examined Nano-Banana2 and report several qualitative results (see anonymous link). It still shows noticeable content leakage in challenging cases where our method handles them well, suggesting that the style-content disentanglement problem has not yet been fully resolved even by closed-source models.
>
> Finally, both $\lambda$ and $\gamma$ are interpretable control variables. $\lambda$ sets the inference-time trade-off between style fidelity and prompt alignment, so Figure 8 (left) reflects an adjustable frontier rather than poor robustness. $\gamma$ prevents degenerate solutions: because CLIP image embeddings are content-biased, we set $\gamma < 1$ to limit the contribution of $y_c$ in reconstruction and force $y_s$ to retain sufficient style details.
>
> # Q3 DiT
> The framework can be generalized to DiT. The core of our method is to distill a style representation from the reference image and inject it alongside the text prompt during generation. Thus, it depends more on whether the backbone provides a suitable conditioning interface than on any particular model structure. We have preliminarily migrated it to a DiT backbone with an image-conditioning interface (FLUX.1-dev) and verified its basic feasibility. Since systematic evaluation is still ongoing, we do not make further claims about final DiT performance at this stage.
>
> # Q4 $L_{orth}$ decoupling
> Stable decoupling in our method is not achieved by $L_{orth}$ alone, but by jointly designed objectives. Anchor loss guides $y_c$ toward content semantics and $y_s$ toward the style direction, orthogonality suppresses their overlap and reconstruction preserves sufficient style information in $y_s$. Thus, $L_{orth}$ mainly reduces overlap and suppresses content leakage, rather than ensuring stability by itself. To further avoid collapse, we set $\gamma < 1$ to limit the contribution of $y_c$ in reconstruction and force $y_s$ to preserve sufficient style details. Moreover, because our method optimizes only the lightweight StyleDistiller with the backbone frozen, its optimization problem is far smaller than full-model personalization, making 300 steps sufficient for stable convergence in the single-reference setting.
>
> # Q5 Style-structure boundary
> Style and structure are not strictly binary opposites in our setting; some cues like line exaggeration naturally lie near their boundary. We therefore adopt a task-oriented operational definition: rendering-related patterns (e.g., color palette, texture, brushwork, lighting, line quality, visual tone) are treated as style, whereas objects, scenes, and the global structure/layout of the reference image are treated as content semantics. Under this definition, content leakage means directly reproducing the reference image’s specific objects or overall structure/layout in the generated result. Such copied structure is often incompatible with the target prompt and appears as structural confusion. By contrast, the few cases in Figure 6 with line-organization changes or shape exaggeration are better understood the transfer of stylistic cues near the style-structure boundary, rather than obvious layout leakage, since they do not clearly replicate the reference image’s main overall structure.

---

> > ### Author Rebuttal · Reviewer_hXvm · 2026-04-03
> >
> > Thank you for the response. The authors' rebuttal has largely addressed my concerns, and I will raise my score accordingly.

---

> > > ### Author Response · Authors · 2026-04-03
> > >
> > > Thank you very much for your follow-up comment and for your careful reconsideration of our paper. We truly appreciate your note that our rebuttal has addressed your concerns.
> > >
> > > We just wanted to kindly note that the score currently visible in the system still appears unchanged on our side, and we were wondering whether the update might simply still be pending. We completely understand that this may just be due to timing, and we would of course be happy to provide any further clarification if helpful.
> > >
> > > Thank you again for your time and thoughtful evaluation.

---

### Official Review · Reviewer_Fdt1 · 2026-03-08

**Soundness:** 3
**Presentation:** 3
**Significance:** 2
**Originality:** 2
**Overall Recommendation:** 4
**Confidence:** 4

**Summary:**

This paper addresses text-guided stylized image generation using diffusion models. The authors first identify the content leakage issue inherent in CLIP embeddings, then clarify that methods removing content descriptions from style descriptions result in loss of style. Against this backdrop, they propose StyleDistiller. The proposed method first generates style descriptions and content descriptions from a style image using MLLM. It then applies cross attention between the embeddings of the two descriptions and the style image embedding. Four losses are applied to the resulting features: 1) Anchor Loss constrains the feature vector of (content description * style image) to approach the feature vector of (content description), while simultaneously constraining the feature vector of (style description * style image) to approach the feature vector of the style image. 2) Orthogonality Loss constrains the feature vector of (content description * style image) to diverge from the feature vector of (style description * style image). 3) Reconstruction Loss constrains the sum of the feature vectors of (content description * style image) and (style description * style image) to approach the feature vector of the (style image). 4) Denoise Loss is the loss used in diffusion models to reconstruct the style image. Furthermore, the authors propose Prompt Alignment Enhancement, which improves fidelity to the text prompt by mixing the text prompt's features into the (style description * style image) features during inference. Experiments evaluate performance using two metrics: how well the style image's style is preserved and fidelity to the text prompt. A user study is also conducted. Comparisons with existing methods reveal that the proposed method outperforms them, particularly in terms of fidelity to the text prompt.

**Compliance With Llm Reviewing Policy:**

Affirmed.

**Final Justification:**

The authors' response has addressed my concerns. If the paper is revised accordingly, I would be inclined to give it a weak accept.

**Key Questions For Authors:**

Please see the Weaknesses.

**Limitations:**

yes

**Strengths And Weaknesses:**

**Strengths**

1. The authors clearly identify the problem that methods removing content descriptions from style descriptions result in style loss. Against this backdrop, they propose a method to remove content features from style features in the feature space. This idea is compelling.

2. The authors propose a loss function that drives style features and content features apart, successfully separating content features from style features. This idea is also compelling.

3. The authors use two metrics in their experiments: how well the style of the style image is preserved and fidelity to the text prompt. They also conduct a user study. Sufficient experiments are performed to demonstrate that the proposed method outperforms existing methods.


**Weaknesses**

1. The explanation of the proposed method is not formal, making it difficult to grasp its content. Specifically, the authors should clarify whether each feature is a vector or a matrix, and its size.

2. Cross attention is computed between $f^s_t$ and $f^s_I$, and between $f^c_t$ and $f^s_I$. This suggests $f^s_t$ and $f^c_t$ each have the length of text tokens, while $f^s_I$ has a length equal to the number of patches. If so, $L_{orth}$ is likely computing inner products between features of different lengths. How this is implemented remains unclear.

3. Recent papers [1,2] discuss issues like content leakage and overfitting to style images. The authors should cite these papers to clarify the differences. Furthermore, these methods should be used for comparison in the experiments.

[1] Zhu, Lin, et al. "Less is More: Masking Elements in Image Condition Features Avoids Content Leakages in Style Transfer Diffusion Models." The Thirteenth International Conference on Learning Representations, 2025.

[2] Lei, Mingkun, et al. "Stylestudio: Text-driven style transfer with selective control of style elements." Proceedings of the Computer Vision and Pattern Recognition Conference. 2025.

---

> ### Author Rebuttal · Authors · 2026-03-31
>
> We thank the reviewer for the feedback and address the concerns below.
>
> # W1 Clarification of feature definitions
> To clarify, we list below whether each feature is a vector or matrix, together with its actual shape in our OpenCLIP ViT-bigG-14 implementation:
>
> - $f_t^s, f_t^c$: vectors, i.e., the global CLIP embeddings of the style/content texts, shape $[1,1280]$.
> - $H_I$: matrix, i.e., the last hidden states of reference images in the CLIP image encoder, shape $[257,1664]$. Before entering cross-attention, it is passed through a learnable linear layer and projected to $[257,1280]$.
> - $y_s, y_c$: vectors, i.e., the final style and auxiliary content representations, shape $[1,1280]$.
> - $g_I$: vector, i.e., the global CLIP image embedding of the reference image used in the loss computation, shape $[1,1280]$.
>
> For simplicity, the paper used the same notation $f_I^s$ for both the global image feature $g_I$ and the intermediate image feature $H_I$, and also omitted the projection from $H_I$ to its cross-attention input. The actual data flow is:
>
> $f_t^s/f_t^c\,[1,1280]$ as query + $H_I$ after projection $[257,1280]$ as key/value $\rightarrow$ cross-attention $\rightarrow [1,1280]$ intermediate feature $\rightarrow$ linear layer $\rightarrow y_s/y_c\,[1,1280]$.
>
> We apologize for the inconvenience in understanding caused by this ambiguity. We will revise the notation accordingly and explicitly add the corresponding shapes and data flow in the paper.
>
> # W2 Clarification of the computation of $L_{orth}$
> We apologize for the unclear presentation. To clarify, $L_{orth}$ is not computed on mismatched-length features, but on $y_s$ and $y_c$, both of shape $[1,1280]$ (see W1). In StyleDistiller, we use the global CLIP features of the style/content texts to aggregate $y_s$ and $y_c$ from the token-level image features, leveraging the overall semantic difference between the style and content texts to extract two corresponding global representations from the reference image. And $L_{orth}$ is designed to encourage $y_c$ to focus on content-related information while pushing $y_s$ to capture complementary style information. Figure 7 also support this interpretation: removing $L_{orth}$ leads to more severe reference content leakage. We will clarify this point in revision.
>
> # W3 More Comparison
> We have added discussion and experimental comparisons with the suggested recent methods, MaskST [1] and StyleStudio [2]. Both methods address content leakage and prompt alignment in text-driven stylized image generation, but differ from ours in both technical path and resulting trade-off. MaskST reduces leakage by masking content-related image-condition features; however, when style and content are strongly entangled, such masking can become less precise and may suppress useful style cues together with content-related ones. StyleStudio improves text alignment through generation-time intermediate-feature control, but such stronger generator-level intervention may sacrifice some style information. In contrast, our method first distills a cleaner and more controllable style representation from a single reference image through an observation-driven formulation, and then uses the lightweight PAE module to improve prompt alignment in a controlled manner. This allows our method to achieve a better balance between style fidelity and prompt alignment, while also naturally supporting style editing and fusion. We further compared the three methods under a unified protocol:
>
> | Method | CSD $\uparrow$ | CLIP-Text $\uparrow$ |
> |---|---:|---:|
> | **Ours** | **53.26** | **28.95** |
> | StyleStudio | 41.77 | 28.06 |
> | MaskST | 48.78 | 26.25 |
>
> The results support the discussion above. MaskST, due to its masking-based suppression, achieves lower CSD and CLIP-Text than ours, while StyleStudio attains relatively strong text alignment but lower style fidelity. In contrast, our method achieves the best results on both metrics, indicating a better overall balance between style fidelity and prompt alignment. Moreover, because our method learns an explicit and controllable style representation, it also naturally supports personalized style manipulation such as style editing and fusion. We will add this discussion in the revision to better clarify the differences between our method and these recent works.

---

> > ### Author Rebuttal · Reviewer_Fdt1 · 2026-04-02
> >
> > I would like to thank the authors for their response.
> >
> > They have clarified the points that were unclear, and they have also demonstrated that the proposed method achieves higher performance than state-of-the-art methods.
> >
> > Based on these explanations, I will adjust the scores upward.

---

> > > ### Author Response · Authors · 2026-04-02
> > >
> > > Thank you again for your thoughtful follow-up and for taking the time to read our rebuttal carefully. We sincerely appreciate your note indicating that our clarifications addressed your concerns and that you were considering adjusting the score upward.
> > >
> > > We just wanted to gently check whether this has already been reflected in the system, as we noticed that the score currently appears unchanged on our side. We completely understand that this may simply be an oversight or a system delay. If there is any additional clarification that would be helpful from us, we would be very happy to provide it.
> > >
> > > Thank you again for your time and consideration.

---

### Official Review · Reviewer_ZVDa · 2026-03-13

**Soundness:** 3
**Presentation:** 3
**Significance:** 2
**Originality:** 3
**Overall Recommendation:** 4
**Confidence:** 4

**Summary:**

This paper introduces StyleDistillation, a framework designed for zero-shot, text-guided stylized image generation and aesthetic manipulation. The authors identify a critical issue in current style transfer methods: the severe entanglement of image content and style, which often leads to content leakage from the reference image.
To address this, they propose a dual-branch architecture that explicitly decouples content from style. The core contribution lies in utilizing an InfoNCE-based contrastive loss to minimize the mutual information between the original image embedding and a text-aligned content embedding. By stripping away the text-aligned semantic components, the framework distills a purified "style embedding," which is then injected into the diffusion process via a cross-attention mechanism. The method enables applications like stylized generation, style editing, and multi-style fusion.

**Compliance With Llm Reviewing Policy:**

Affirmed.

**Final Justification:**

Thanks for the rebuttal. Most my concerns have been solved, so I keep my score.

**Key Questions For Authors:**

**Please note that while I am highly familiar with generative models and latent space control, I am not an expert in the specific sub-field of Style Transfer.** I remain open-minded.
1. Given the 150-second optimization cost per image, how do you justify comparing your approach directly with zero-shot feed-forward methods? Have you compared it with other optimization-based personalization techniques (e.g., Textual Inversion or single-image LoRA)?
2. Why do all compared baselines date back to 2024 or earlier? Could you provide comparisons with the latest 2025 style transfer methods to establish a true SOTA performance?
3. Since the CQ Score relies on population standard deviations and is highly sensitive to the pool of compared methods, can you provide a more robust, absolute metric (such as a Pareto frontier plot or a standard harmonic mean) to demonstrate your method's trade-off advantage?
4. Could you provide a theoretical explanation or feature space visualization (e.g., t-SNE) proving that adding the text embedding directly back into the style embedding (Equation 6) does not corrupt the decoupled style space and reintroduce content entanglement?

**Limitations:**

The authors should discuss the limitations regarding time and computational costs. The 150-second preprocessing optimization time per style image severely restricts the framework's deployment potential in industrial applications requiring high concurrency or real-time style transfer (e.g., real-time video filters, instant image generators).

**Strengths And Weaknesses:**

### Strengths

* Originality: The empirical analysis of style characteristics in the CLIP embedding space is solid. Using MLLM-generated content descriptions as soft semantic anchors and extracting pure style features via orthogonal constraints is elegant and logically sound.

* Presentation: The generated results are impressive. Compared to existing methods, StyleDistillation faithfully preserves complex aesthetic features of the reference image while adhering strictly to the text prompt.

* Versatility: The distilled high-purity style representations seamlessly support complex downstream aesthetic operations, such as multi-style fusion and instruction-based fine-grained style editing, greatly enhancing its practical value.


### Weaknesses

* Evaluation: **1.** The proposed StyleDistiller requires 300 optimization steps, taking 150 seconds on an RTX-4090 per reference image. This is essentially a Test-time Optimization method. However, Table 1 directly compares it with zero-shot, feed-forward methods like IP-Adapter, CSGO, and StyleShot, which only take milliseconds. Comparing generation quality directly without acknowledging this massive disparity in computational cost is unfair. **2.** The comparative analysis heavily relies on methods published in 2024 or earlier. For an ICML 2026 submission, the lack of comparisons with the latest 2025 SOTA style transfer methods weakens the claim of having a leading edge.
* Soundness: **1.** To evaluate the trade-off between style and content, the authors introduce the Composite Quality (CQ) Score. However, the denominator uses the standard deviations ($\sigma_{sty}, \sigma_{text}$) of the specific methods chosen for comparison. This makes the CQ Score an extremely unstable relative metric. Introducing or removing an extreme baseline would cause drastic fluctuations in the standard deviation, potentially overturning the rankings of all methods. **2.** The core contribution of the method relies on complex training objectives (e.g., $\mathcal{L}_{orth}$) to strictly decouple style and textual content. Yet, in the inference stage, the Prompt Alignment Enhancement (PAE) module uses a simple linear interpolation ($y_s^p = y_s + \lambda \cdot f_t^p$) to add the text prompt embedding directly back into the style embedding (Equation 6). Intuitively, this simple linear addition re-entangles semantic content back into the style representation, logically undermining the purity achieved during the distillation phase.

---

> ### Author Rebuttal · Authors · 2026-03-31
>
> We thank the reviewer for the feedback. Additional tables and results: https://anonymous.4open.science/r/StyleDistillation-D4AE/ZVDa.md
>
> # Q1. Comparison scope
> Compared with zero-shot feed-forward methods, our method follows a test-time optimization paradigm. Our comparisons are based on task consistency, i.e., evaluating different methods under the unified setting of single-reference, text-guided stylized image generation, to provide a comprehensive assessment of their performance rather than claim strict alignment in computational budget. For this task, training style adapters from large-scale style datasets is an important technical route, so comparisons with methods like CSGO are necessary.
>
> We also compare with other technical routes, including training-free methods such as StyleAligned and textual inversion methods such as DreamStyler. To further address the concern, we also include UnZipLoRA [1], a single-image LoRA method, and our method still performs better (see anonymous link).
>
> This does not mean we ignore the extra 150s cost. We include representative methods from different routes for a more complete evaluation. Among optimization-based methods, ours is both better and faster than DreamStyler (\~720s) and UnZipLoRA (\~1200s). Although it introduces an upfront cost relative to zero-shot methods, it also brings better performance; moreover, this optimization is performed only once per reference style and can be amortized across multiple generations. The resulting stable style representation also supports style fusion and style editing. Overall, our method achieves an optimal balance between model performance and practical deployment cost.
>
> # Q2. More Comparison
> We additionally campare two recent 2025 methods, StyleStudio[2] and MaskST[3] (see anonymous link). Our method outperforms both on CSD and CLIP-Text, indicating that the proposed method remains competitive.
>
> # Q3. CQ metric
>
> CQ is introduced as a relative summary metric for jointly reflecting CSD and CLIP-Text, rather than the sole evidence for the style fidelity–prompt alignment trade-off. To further provide this, we followed the reviewer's suggestion to compute the standard harmonic mean of the original CSD/CLIP-Text values in Table 1, and also plot the Pareto frontier (see anonymous link). Our method still ranks first in harmonic mean and lies on the Pareto frontier. Moreover, except for IP-Adapter, all other methods are dominated by ours; and as already reflected in the paper, IP-Adapter mainly suffers from the lowest CLIP-Text among all methods. Therefore, our conclusion that the proposed method achieves a favorable style fidelity--prompt alignment trade-off is not supported solely by CQ.
>
> # Q4. PAE
> We first clarify the roles of different components in our method, and then provide the theoretical motivation for PAE. The upstream StyleDistiller removes irrelevant content information and extracts a high-quality style representation. In contrast, PAE is a lightweight inference-time module whose role is not to further purify style, but to use the disentangled style representation to better balance style fidelity and prompt alignment.
>
> Its motivation is that when the style representation $y_s$ and the text prompt are both used as conditions, the model tends to prioritize $y_s$, which carries richer visual cues. We therefore construct a fused representation $y_s^p$ to improve prompt alignment. Since Observation 2 treats text as soft semantic guidance rather than an exact content component, we minimize the objective function $\min_{y_s^p}\frac{1}{2}\|y_s^p-y_s\|_2^2-\lambda\langle y_s^p,f_t^p\rangle$, where the first term keeps $y_s^p$ close to $y_s$, and second encourages consistency with the text prompt. Its closed-form solution is Eq. (6). In this sense, Eq. (6) is a prompt-aware correction with controllable strength. This provides a theoretical motivation at the representation level.
>
> Our experiments support this interpretation. Figure 8 shows that as $\lambda$ increases, prompt alignment improves while style similarity gradually decreases, indicating that PAE introduces an interpretable and controllable trade-off rather than uncontrolled damage to the style space. Meanwhile, Figure 7 shows that significant content leakage mainly appears when the upstream disentanglement constraints are removed, rather than being caused by Eq. (6) itself. We also tried other design like AdaIN-based intermediate feature fusion, but despite higher inference cost it gave worse text alignment than PAE (27.52 vs. 28.95). We therefore adopted Eq. (6) because it provides effective prompt-alignment enhancement with minimal complexity.
>
> [1] UnZipLoRA: Separating Content and Style from a Single Image. ICCV 2025
>
> [2] StyleStudio: Text-driven Style Transfer with Selective Control of Style Elements. CVPR 2025
>
> [3] Less is More: Masking Elements in Image Condition Features Avoids Content Leakages in Style Transfer Diffusion Models. ICLR 2025

---

> > ### Author Rebuttal · Reviewer_ZVDa · 2026-04-02
> >
> > The responses are mostly acceptable. Although I still have slight subjective reservations regarding Q4, I have no further questions.

---

> > > ### Author Response · Authors · 2026-04-04
> > >
> > > Thank you very much for your thoughtful follow-up and for your positive evaluation of our rebuttal. We are truly grateful for your acknowledgment that our response has adequately addressed your concerns, and we sincerely appreciate your careful, open-minded, and constructive assessment of our work.
> > >
> > > If convenient, we would be very grateful for any further clarification on the slight remaining concern from your perspective. If there is any point that you feel is not yet fully resolved, we would sincerely appreciate the opportunity to understand it more clearly and address it as best we can. Should this remaining point be clarified to your satisfaction, we would be grateful if you might take this into account in your final evaluation.
> > >
> > > Thank you again for your time and consideration.

---

### Official Review · Reviewer_1mqT · 2026-03-13

**Soundness:** 3
**Presentation:** 3
**Significance:** 2
**Originality:** 3
**Overall Recommendation:** 4
**Confidence:** 4

**Summary:**

This paper proposes StyleDistillation, a framework for single-reference, text-guided stylized image generation. The core idea is to distill a pure style representation from a reference image's CLIP embedding by leveraging two empirical observations about the CLIP space: (1) style offsets are geometrically consistent across different contents, and (2) textual descriptions are better suited as soft semantic guidance rather than exact content components for subtraction. Based on these observations, the authors design a lightweight StyleDistiller module with two parallel branches (style and auxiliary content), trained with a combination of anchor, orthogonality, reconstruction, and denoising losses. At inference time, a Prompt Alignment Enhancement (PAE) mechanism adds a scaled text prompt embedding to the style representation to improve content alignment. The method also supports style editing (adding/removing style elements via text) and style fusion (combining multiple style representations). Experiments on 40 reference styles and 30 prompts show competitive performance on CSD (style fidelity) and CLIP-Text (prompt alignment) scores, with strong user study results.

**Compliance With Llm Reviewing Policy:**

Affirmed.

**Final Justification:**

Thanks for the rebuttal. My concerns have been solved, so I raise my score.

**Key Questions For Authors:**

See weakness.

**Limitations:**

Yes.

**Strengths And Weaknesses:**

- The two observations about CLIP space are empirically grounded using the UnlearnCanvas dataset with concrete metrics (silhouette scores, nearest-neighbor analysis). The finding that style offsets cluster after removing content projections is insightful and well-supported.
- The paper is generally well-written and clearly structured.
- The style editing capability (adding/removing style elements via text) is a genuinely useful application.

**Weakness:**
- The PAE mechanism (Equation 6) is essentially a simple additive combination of the style embedding and text prompt embedding with a fixed scalar $\lambda$. This is somewhat ad hoc and lacks theoretical justification for why simple addition in CLIP space should yield a balanced trade-off. The method's sensitivity to $\lambda$ (shown in Figure 8) suggests this balance is fragile.
- The claim that Observation 1 constitutes a "geometric prior" is somewhat overstated. The de-projection procedure used to reveal style clusters requires knowing the content grouping, which is unavailable in the single-reference setting. The connection between the dataset-level observation and the per-image optimization is indirect.
- The notation is occasionally inconsistent or overloaded. For instance, $f_I^s$ uses superscript $s$ to denote the style reference image, but $f_t^s$ and $f_t^c$ use superscripts for style/content descriptions, which can be confusing.
- The overall pipeline (CLIP image encoder → lightweight module → IP-Adapter injection into SDXL) follows a well-established pattern. The PAE mechanism and style editing/fusion operations are conceptually straightforward (additive operations in embedding space).
- The comparison in Table 1 shows that IP-Adapter achieves a CSD score of 78.04, substantially higher than your method's 53.26. While the authors argue this comes at the cost of poor text alignment. I suggest the authors provide a more detailed analysis of when the proposed method's style fidelity falls short.
- The proposed method was only compared with open-sourced models. I wonder if the tasks shown in the paper are also difficult for business models such as nano-banana or Seed. At least some bad cases of the business models should be shown.

---

> ### Author Rebuttal · Authors · 2026-03-31
>
> We thank the reviewer for the feedback and address the concerns below. Supplementary tables and qualitative results are provided here:
>
> https://anonymous.4open.science/r/StyleDistillation-D4AE/1mqT.md
>
> # W1.PAE/$\lambda$
> We would like to clarify that Eq. (6), as a lightweight inference-time design, is not merely a heuristic scheme. Its motivation is that when the style representation $y_s$ and the text prompt are both used as conditions, the model tends to prioritize $y_s$, which carries richer visual cues. We therefore construct a fused representation $y_s^p$ to improve prompt alignment.
>
> Since Observation 2 treats text as soft semantic guidance rather than an exact content component, we minimize the objective function $\min_{y_s^p}\frac{1}{2}\|y_s^p-y_s\|_2^2-\lambda\langle y_s^p,f_t^p\rangle$, where the first term keeps $y_s^p$ close to $y_s$, and second encourages consistency with the text prompt. Its closed-form solution is Eq. (6). Thus, Eq. (6) is better understood as a regularized trade-off between preserving style and enhancing prompt alignment, rather than a purely empirical combination. This provides a theoretical motivation at the representation level.
>
> We also view $\lambda$ as a controllable trade-off.  Larger $\lambda$ improves prompt alignment but may move the output away from the reference style; thus Fig.8 shows an interpretable movement along the style/prompt frontier rather than a lack of robustness. We also tried more complex fusion, e.g., AdaIN-based intermediate feature fusion, but despite higher inference cost it gave worse text alignment than PAE (27.52 vs. 28.95). We therefore adopted Eq. (6) because it provides effective prompt-alignment enhancement with minimal complexity.
>
> # W2. Geometric prior
> We thank the reviewer for this suggestion, which helps make our wording more precise. More accurately, Observation 1 is a dataset-level empirical finding: after suppressing the content direction, style exhibits cross-content geometric consistency in CLIP space. It motivate our optimization design: once content interference is suppressed through Observation 2 together with the orthogonality/reconstruction constraints, minimizing
> $\|y_s-f_I^s\|_2^2$
> serves as a reasonable surrogate signal for learning the style representation. The main experiments already support this design.
>
> To further address this concern, we sampled one reference image per style from UnlearnCanvas (60 styles total) and reran all baselines. Our method still achieves the best overall performance (see anonymous link). This result further strengthens its empirical connection to our method's effectiveness. We will revise this wording in the revision.
>
> # W3. Notation
> We have revised the notation to make it consistent, especially the superscript/subscript conventions, and clarified this in the main text and symbol table.
>
> # W4. Pipeline
> We adopt a standard pipeline by design, allowing us to focus on the core contribution: distilling from a single reference image a style representation for generation and manipulation. We analyze the CLIP space from the style perspective and design collaborative optimization objectives based on two key observations. Although PAE and style editing/fusion operate at the embedding level, they are not detached heuristic add-ons: their feasibility relies on the upstream insight-driven style representation distillation and thus demonstrate the effectiveness and manipulability of the learned representation.
>
> # W5. Lower CSD
> Although IP-Adapter achieves a higher CSD, this comes at the cost of much worse text alignment (the lowest CLIP-Text among all methods). Our method instead aims for a better balance, as jointly evidenced by the CLIP-Text and CQ results. As a deliberate trade-off, it does not pursue a high CSD at the expense of prompt alignment when style is highly entangled with content/local semantics. In such cases, it suppresses these content-entangled cues (e.g., via the orthogonality constraint) to reduce leakage and preserve prompt alignment. This trade-off is often not rewarded by CSD, although our qualitative results and user study suggest better overall outputs. Consistently, Fig. 7 shows that removing the orthogonality constraint increases CSD but also causes more reference content leakage. We further found that strong structural perturbations yield a CSD of 86.93 to the original images, even higher than the average CSD between different images of the same style (71.13), suggesting that CSD over-rewards local texture/color similarity while often underreacting to global structural disorder caused by severe content leakage in methods such as IP-Adapter.
>
> # W6. Closed-source models
>
> We additionally evaluated Nano-Banana2 (see anonymous link for representative bad cases). It still shows noticeable content leakage in challenging cases where our method handles them well, suggesting that the style-content disentanglement problem has not yet been fully resolved even by closed-source models.

---

> > ### Author Rebuttal · Reviewer_1mqT · 2026-04-02
> >
> > Thanks for the rebuttal. My concerns have been solved, so I raise my score.

---

### Decision · Program_Chairs · 2026-04-30

**Decision:**

Accept (regular)

**Comment:**

This paper was reviewed by 4 experts in the field. After discussion, the reviewers still hold a consistent review to this work. The rating is 4(weak accept), 4(weak accept), 4(weak accept), 4(weak accept).

In general, all reviewers agree that this work effectively separates style and content using a decoupling formulation and specific loss functions. Still, reviewers raised several concerns to this work. The concern includes 1) the framework is viewed as an incremental improvement over existing architectures, and 2) comparisons with zero-shot methods may be unfair due to higher computational costs. Particularly, area chair also agreed tha the comparison is not fair and authors shall point this out in the revision.

Still, given the novel design, the decision of this work is to Accept.  We strongly recommend the authors carefully read all reviewers’ final feedback and revise the manuscript as suggested in the final camera-ready version if being accept.